# Strong sesquiterpene emissions from Amazonian soils

E. Bourtsoukidis [1], T. Behrendt[2], A.M. Yañez-Serrano [1,3,8], H. Hellén[4], E. Diamantopoulos[5], E. Catão[2],
K. Ashworth[6], A. Pozzer [1], C.A. Quesada[3], D.L. Martins[3,9], M. Sá[3], A. Araujo[3], J. Brito [7,10], P. Artaxo[7],
J. Kesselmeier [1], J. Lelieveld [1] & J. Williams[1]

The Amazon rainforest is the world's largest source of reactive volatile isoprenoids to the atmosphere. It is generally assumed that these emissions are products of photosynthetically driven secondary metabolism and released from the rainforest canopy from where they influence the oxidative capacity of the atmosphere. However, recent measurements indicate that further sources of volatiles are present. Here we show that soil microorganisms are a strong, unaccounted source of highly reactive and previously unreported sesquiterpenes ($C_{15}H_{24}$; SQT). The emission rate and chemical speciation of soil SQTs were determined as a function of soil moisture, oxygen, and rRNA transcript abundance in the laboratory. Based on these results, a model was developed to predict soil–atmosphere SQT fluxes. It was found SQT emissions from a Terra Firme soil in the dry season were in comparable magnitude to current global model canopy emissions, establishing an important ecological connection between soil microbes and atmospherically relevant SQTs.

[1] Atmospheric Chemistry and Biogeochemistry Departments, Max Planck Institute for Chemistry, Hahn-Meitner-Weg 1, 55128 Mainz, Germany. [2] Max Planck Institute for Biogeochemistry, Hans-Knöll-Straße 10, 07745 Jena, Germany. [3] National Institute of Amazonian Research, Av. André Araújo, 2936 - Petrópolis, Manaus, AM 69067-375, Brazil. [4] Finnish Meteorological Institute, Erik Palménin aukio 1, FI-00560 Helsinki, Finland. [5] Department of Plant and Environmental Science, University of Copenhagen, Thorvaldsensvej 40, 1871 Frederiksberg C, DK-1871 Copenhagen, Denmark. [6] Lancaster Environment Centre, Lancaster University, Lancaster LA1 4YQ, UK. [7] University of Sao Paulo, Rua do Matão, Travessa R, 187, São Paulo, SP CEP 05508-900, Brazil. [8] Present address: University of Freiburg, Georges-Köhler-Allee 53, 79110 Freiburg, Germany. [9] Present address: Imperial College London, London SW7 2AZ, UK. [10] Present address: Laboratory of Atmospheric Physics (LaMP), University Blaise Pascal, 63000 Clermont-Ferrand, France. Correspondence and requests for materials should be addressed to E.B. (email: e.bourtsoukidis@mpic.de)

Sesquiterpenes are a chemically diverse class of volatile isoprenoids relevant to biology, ecology, and due to their high reactivity to ozone and prodigious particle production efficiency to atmospheric composition[1–5]. They are known to be emitted to the air from plants as a function of oxidative and thermal stress[6–9]. By generating strong spatial gradients, due to their rapid reaction with ozone, they may affect olfactory navigation in pollinating insects[10].

The substantial diversity of species within this compound class, the rapid reaction with ozone, the low ambient mixing ratios, and the low volatility make it difficult to quantify them accurately and to elucidate impacts within the ecosystem[8,11,12]. Current flux parameterizations are formulated based on scarce measurements from plant systems only, unified with an empirical temperature response for all the species in the family[13,14]. Although some sesquiterpenes have been detected previously in soil[15–18], the relevance of this source to the atmosphere is unclear. In particular, carbon-rich tropical soils, which have not been examined previously[19], could be potent but overlooked sources.

In this study, we use a laboratory-derived emission algorithm to evaluate the impact of Amazonian soils on the net ecosystem SQT flux and hence atmospheric chemistry. We combine proton transfer–mass spectrometry (PTR-MS) and gas chromatography–mass spectrometry (GC-MS) methods to evaluate the soil SQT emission strength in the laboratory and in the field. The laboratory incubations reveal strong emissions of SQTs from soils as a function of water-filled pore space (WFPS), allowing the development of an emission algorithm that was validated with field samples. Simulated results compared closely with SQT flux measurements in the field, so a two-year period (2014–2015), was modeled based on in situ rainfall and soil moisture measurements. The simulations indicate that SQT emissions from soils are in comparable magnitude with canopy emissions while they dominate $O_3$ reactivity in the forest floor.

## Results

**Laboratory observations.** To quantify and chemically speciate SQT emission fluxes, 42 soil samples were collected at three depths from eight Amazonian sites (Supplementary Table 1). The samples included Ferralsols, Alisols, and Podzols, originating from three major ecosystems: dense Terra Firme forest (TF), floodplain terrace (FLT), and white sand (WS). While most of TF soils investigated are in the pristine Amazonian forest in the vicinity of the Amazonian tall tower observatory (ATTO), TF4 and TF5 are located at the ecotone of rainforest to cerrado (tropical savanna ecoregion) and were part of the large-scale, long-term fire experiment[20]. The soils were selected on the basis of type to represent the majority of soils in the Amazon basin[21]. In all laboratory-based experiments, the soil atmosphere was simulated by mixing environmentally relevant VOC ratios and $CO_2$ either into zero air or pressurized $N_2$ to simulate aerobic and anaerobic conditions, respectively. At the beginning of each experiment, the soils were wetted to 100% water-filled pore space (WFPS) and allowed to desiccate in a controlled and continuously monitored environment[22].

Upon the initial wetting, a strong burst release of SQTs was observed for the majority (>80 %) of the incubated soils from TF and FLT but not from WS. Following the initial burst directly after wetting, soil VOC emissions in the laboratory stabilized and 2–3 days later SQTs ($C_{15}H_{24}$) and acetone ($C_3H_6O$) displayed clear and reproducible optima as a function of soil moisture (Fig. 1). Such emission optima have been previously linked to microbial activity for NO and HONO[23,24] over similar moisture ranges (WFPS$_{SQT,opt.}$ = 31.3 ± 6.3%, WFPS$_{acetone,opt.}$ = 10.2 ± 2.7%). Acetone emissions were the strongest, with release rates

up to 1.5 mg m$^{-2}$ h$^{-1}$; an order of magnitude higher than acetone net fluxes measured above an Amazonian forest[25]. However, the WFPS over which the acetone emissions occur is significantly lower than conditions normally experienced in the Amazonian rainforest. Contrasting behavior was reproducibly exhibited by methanol, which was weakly up-taken under wet conditions and released during drying, while monoterpenes were (in contrast

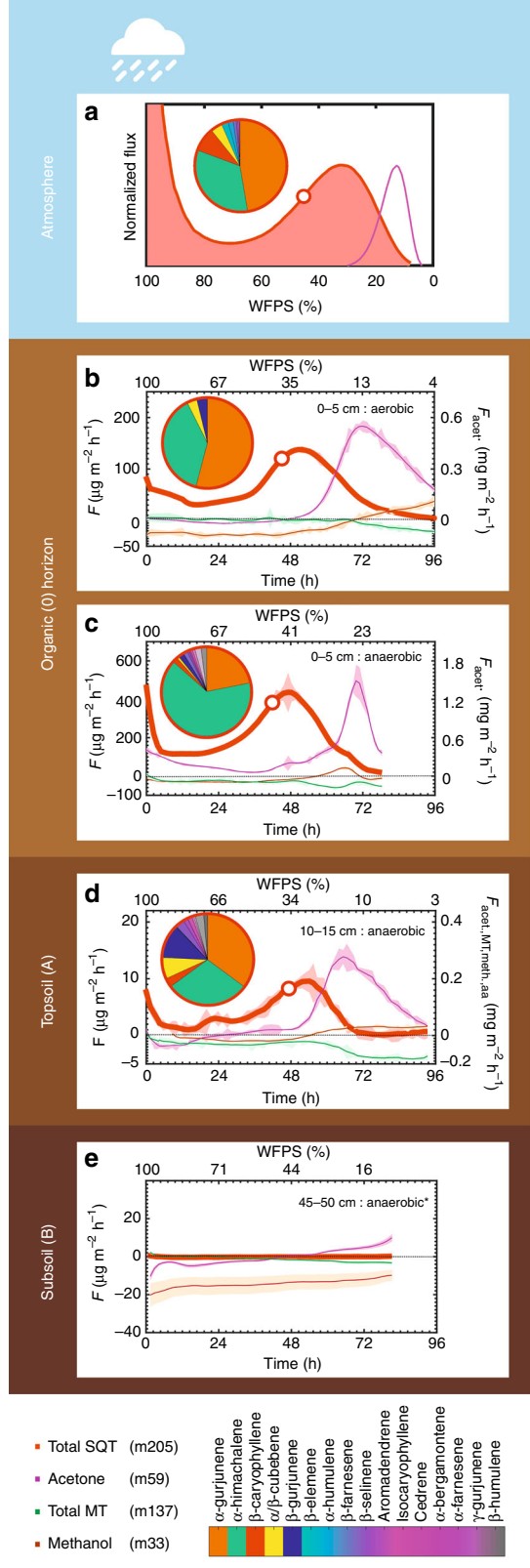

with the canopy fluxes[25]) weakly emitted under wet conditions and moderately consumed in the low-moisture range.

Besides soil moisture, which consistently produced an emission optimum, the most important environmental factors driving SQT production and release were found to be the soil type, depth, and oxygen availability. Temperature only weakly increased SQT emission, as has been noted previously for Mediterranean soils[26], while varying $CO_2$ abundance (400–5000 ppm$_v$) had no effect on the emission pattern or strength. Substantial differences in SQT release were observed from the different sub-ecosystems and soil horizons (Figs. 2 and 3). An order of magnitude stronger emission rate was measured for the TF1 ecosystem. Terra Firme soils (TF1, TF2, and TF3) have very similar physiochemical properties (pH, bulk density, total reserve bases ($\Sigma_{RB}$), clay content) and similar morphological properties[27]. Such a substantial difference in the SQT emission rates between TF1 and the other two Terra Firme sites may be attributable to a higher microbial activity.

To investigate a link between microbial activity and SQT emissions, subsamples of soil were collected during soil desiccation experiments from TF4 to TF5 and 16S- to 18S rRNA transcript abundances (indicator for bacterial and fungal activity, respectively) were quantified at three points: (1) upon wetting, (2) during the optimum, and (3) under dry conditions. TF4 and TF5 belong to the southern part of the Amazonian rainforest and have been shown to be the least fertile but are widespread across eastern Amazonia[28] and were subject to a long-term fire experiment[20]. TF4 is located in the natural forest (control area) and TF5 in the area that was burned every 3 years (B3Yr) during the experiment (2004–2010). As shown in Fig. 4, SQT emissions were reproducibly high for TF4 and 16S rRNA transcript abundance displayed an optimum at similar soil moisture as the maximum of SQT emissions. In contrast, TF5 (burned site) showed negligible emissions of SQTs and the respective 16S rRNA transcript abundance was up to three orders of magnitude less and invariant over all samples. The 18S rRNA transcript abundance was equally high during the optimum for both soils, indicating that the role of fungal emissions is insignificant relative to bacterial SQT emissions for these soils. Despite the limitations that may arise from the use of rRNA as an indicator of microbial activity[29], both the ecological history and rRNA dynamics observed indicate that the microbial activity drive the SQT production and release for these soils. TF4 and TF5 are typical for a drier region of the Amazonian basin and in addition, SQT emissions from fungi are strongly dependent on fungal-age, rather than biomass[30]. Therefore, the bacterial/fungal contribution to the net SQT production in the Amazon basin requires

further investigation, particularly as a function of season and fungal development stage.

In the absence of oxygen, stronger VOC emissions are expected[31]. This is indeed the case for SQT emissions from the organic (O) horizon (Fig. 3). Along with the production strength, the chemical diversity of SQT emissions increased (see Fig. 1c and Supplementary Fig. 1). While only four SQT species were measured from TF1 under aerobic conditions (α- and β-gurjunene, α-himachalene, and α/β-cubebene), in the absence of oxygen, a total of ten different SQTs were released and the emission ratios have changed markedly. A correspondingly broad spectrum of SQT was identified from the topsoil (A) horizon (10–15 cm). Field samples at TF1 contained the species that were seen only under anaerobic conditions in the laboratory (see Fig. 1a and Table 1). This is a clear indication that the uppermost aerobic organic (O) horizon is not the only source for the SQTs entering the atmosphere, and that the subsurface layer contributes to the emission flux since SQTs can rapidly travel up through soil with very small losses[32].

**Emission algorithm.** The emissions observed in the laboratory are the result of conditions commonly occurring in nature. A natural rain event initiates a cascade of physiochemical and microbial processes as the water percolates through the soil layers. After the rain, the SQT emission burst declines exponentially and then stabilizes as the optimum-shaped microbial emissions start to increase. Therefore, the emission dynamics could be divided into two distinct ranges of soil moisture: a high moisture regime (HM; see cyan area in Fig. 5a, b) up to ≈80% WFPS and a moderate moisture regime (MM; beige area) from ≈80% WFPS to complete desiccation. We combined these two processes into a single equation that can be used for the quantification of SQT soil-to-atmosphere fluxes in all environments and under both aerobic and anaerobic conditions (Fig. 5a, b). The emission model algorithm was applied in all laboratory experiments, where an emission optimum was observed for a particular WFPS. Close agreement between the emission algorithm and measured emissions ($0.89 < R^2 < 0.97$) indicates that the algorithm can reliably simulate the observed SQT emissions (see example in Fig. 5a, b).

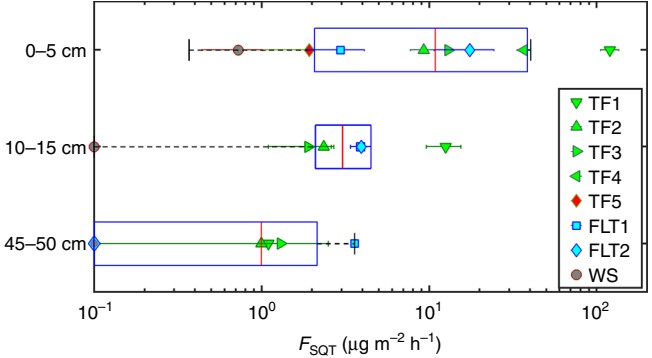

**Fig. 2** Depth profile of optimum SQT emissions. Plotted are SQT flux optimum from all experiments (excluded the points for anaerobic conditions for 0–5 cm; S5, S8, S11, S12) as a function of depth (horizontal box and whisker plots). The markers indicate the values of individual Pits denoted in the figure's legend. The error bar of each marker indicates the standard deviation of replicates (see Supplementary Table 1). SQT emissions from the organic (O) horizon (0–5 cm) were stronger than both Topsoil (A) (10–15 cm) and Subsoil (B) (45–50 cm) combined. TF1 has one order of magnitude stronger emissions than the other TF soil samples and the FLT. Soils that were collected from the middle horizon (10–15 cm) were insignificantly stronger than the bottom soils, apart from TF1

**Fig. 1** Vertical profile of VOC fluxes from a Terra Firme soil (TF1). **a** Normalized soil fluxes of sesquiterpenes (SQTs) (red) and acetone (purple) as a function of water-filled pore space (WFPS). The normalized (to the optimum emission observed at moderate moisture) algorithm-derived emission curve is the result of integrating laboratory observations from **b** organic horizon under aerobic and **c** anaerobic conditions, **d** topsoil (10–15 cm) and **e** subsoil (45–50 cm). SQTs, monoterpenes (MTs), acetone, and methanol are illustrated with the colored lines. Acetone, methanol, and α-pinene were included in the fumigation standard (see Methods). The shaded areas indicate the standard deviation of the measured emission rates at each chamber cycle. The pie charts illustrate the chemical speciation of SQT at the point in time (and WFPS), indicated by the white circle over the SQT measurements (red line). For the color scale, see lower right panel. The pie chart at **a** illustrates the chemical composition under field conditions. The asterisk in **e** denotes that the experiments were performed under aerobic conditions despite the predominant anaerobic conditions in the deep soil layers

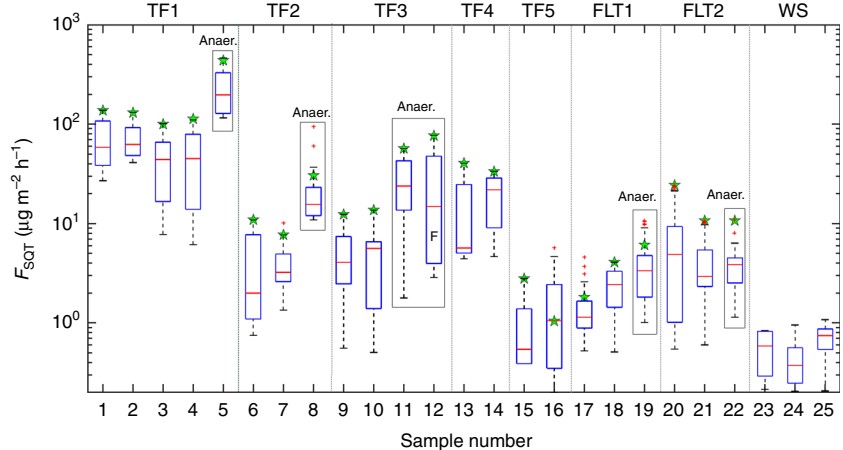

**Fig. 3** Organic (O) horizon (0–5cm) SQT emissions during a complete dry out. All experiments started from saturated moisture conditions. The data represent the emission rates measured with WFPS >10% apart from the samples S5, S8, S11, and S12 that dried up to WFPS ≈ 45%. On each box, the center line (red) indicates the median, and the bottom and top borders of the box (blue) indicate the 25th and 75th percentiles, respectively. The whiskers extend to the most extreme data points not considered outliers, and the outliers are plotted individually using the "+" symbol (red). The boxplot draws points as outliers if they are greater than q3 + w × (q3 – q1) or less than q1 – w × (q3 – q1). q1 and q3 are the 25th and 75th percentiles of the sample data, respectively. The "Whisker" (w) corresponds to ±2.7σ and 99.3% coverage if the data are normally distributed. The green markers indicate the optimum SQT flux (in $\mu g\ m^{-2}\ h^{-1}$). The large boxes around S5, S8, S11, S19, and S22 indicate that these experiments were conducted under anaerobic conditions. The symbol F indicates that sample S12 was flooded to three times its water holding capacity and allowed to dry under zero air as dilution gas

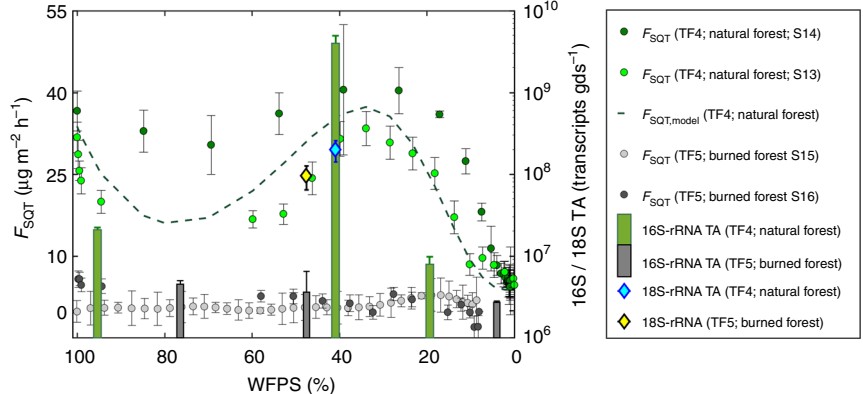

**Fig. 4** SQT emissions and rRNA transcript abundance. SQT emissions (markers, dashed line as model output) and 16S-/18S- rRNA transcript abundance (TA) during desiccation of soil samples from organic (O) horizon from TF4 to TF5. Green circles (dark green for S14 and light green for S15) indicate the SQT fluxes for TF4 while light (S15) and dark gray bullets (S16) indicate the fluxes for TF5. The error bars indicate the standard deviations of four sample points at the respective moisture level. Green bars indicate the 16S rRNA transcript abundance (transcripts per grams of dry soil, $gds^{-1}$) for TF4 (S13) and black/gray bars the transcript abundance ($gds^{-1}$) for TF5 (S16). The cyan diamond indicates the 18S rRNA copies for TF4 (S13) and the yellow diamond the respective result for TF5 (S16). Under wet and dry conditions the 18S rRNA transcripts $gds^{-1}$ were below 1e7 transcripts $gds^{-1}$ and are not displayed. The error bars in the box-whisker plots indicate the standard deviation of four replicates

**Field measurements**. To evaluate the algorithm, flux measurements were made in the field using Teflon chambers placed directly on the surface soil at TF1 and TF3 (ATTO site). Volumetric soil water content is continuously monitored at the ATTO site with six sensors, arranged vertically from 10 to 100 cm depth. Hydrological modeling and in situ volumetric moisture measurements were combined to derive the WFPS in the field. In our model for SQT emissions from Amazonian soils, we use the −3 and −10 cm WFPS to predict field emissions by integrating the emission burst from anaerobic (O) horizon and the microbial emissions from both aerobic (O) and anaerobic (A) horizons (Fig. 5c). Our model considers anaerobic conditions for the first few hours after a strong rainfall event under the HM regime. It has been shown that it is difficult to predict the O$_2$ availability

after rainfall since soil O$_2$ is not a direct function of rain water[33]. Our field measurements quantified exceptionally strong emissions of SQT, 6 h after strong rainfall (25.1 mm in 2 h). These emissions were stronger than our model prediction, indicating that either the emission burst could be stronger compared with the laboratory observations or that the topsoil (A) significantly contributes to anaerobic emission burst. We note that according to our hydrological model, topsoil (A) is very rarely under anaerobic conditions and hence such conditions were not included in the emission model. The chemical speciation of the first sample was very different from the following samples along the natural desiccation process with bergamontene and isocaryophyllene comprising more than 90% of the total SQT detected up to 6 h after the rainfall. The rest of the samples contained a mixture of

**Table 1 Chemical speciation (%) of SQT emissions from Amazonian soils**

| SQT speciation (%) | | | TF1 | | | TF2 | TF3 | | | FLT1 |
| --- | --- | --- | --- | --- | --- | --- | --- | --- | --- | --- |
| | | | LAB | | FIELD | LAB | LAB | | FIELD | LAB |
| SQT name | CAS | RI (NIST) | 0–5 cm | 10–15 cm | Chamber | 0–5 cm | 0–5 cm | 10–15 cm | Chamber | 0–5 cm |
| | | | N = 9 | N = 2 | N = 2 | N = 2 | N = 10 | N = 1 | N = 17 | N = 3 |
| β-caryophyllene | 87-44-5 | | 0.9 (0.9) | 3.4 (0.2) | 10.4 (3.2) | 3 (0.4) | 40.4 (17.4) | 3.1 | 60.5 (38.4) | 4.5 (3.6) |
| aromadendrene | 489-39-4 | | 0.3 (0.3) | 1.6 (0.03) | 0.4 (0.2) | 1.4 (1.4) | n.d. | 1.1 | n.d. | 0.2 (0.2) |
| α-humulene | 6753-98-6 | | n.d. | 0.2 (0.01) | 2.1 (0.6) | 1.7 (0.4) | 0.7 (1.2) | 0.3 | 14.5 (28) | 0.5 (0.7) |
| α-gurjunene | 489-40-7 | | 37.7 (13.5) | 43.6 (8.5) | 42.5 (8.4) | 11.5 (4.9) | 0.5 (1) | 36.4 | 2.7 (3.8) | 9.2 (11) |
| b-farnesene | 18794-84-8 | | n.d. | n.d. | 1 (0.1) | n.d. | n.d. | n.d. | 14.8 (32) | n.d. |
| Isocaryophyllene | 118-54-0 | | n.d. | n.d. | 0.2 (0.2) | n.d. | n.d. | n.d. | 1.8 (7.1) | n.d. |
| β-cubebene | 13744-15-5 | 1381 | 2.1 (1.2) | 5.9 (1.7) | 3.1 (1.8) | 17 (2.7) | 27 (13.5) | 7.3. | 1.5 (2.1) | 27 (18.6) |
| β-elemene | 515-13-9 | 1392 | n.d. | n.d. | 1.7 (1.3) | n.d. | n.d. | n.d. | n.d. | n.d. |
| α-bergamotene | 17699-05-7 | 1407 | 0.2 (0.3) | 0.3 (0.3) | n.d. | 3 (0.9) | 5.9 (5.4) | n.d. | 3.9 (15.8) | 0.5 (0.7) |
| α-cedrene | 469-61-4 | 1415 | 0.3 (0.3) | 1.3 (0.3) | n.d. | 1.3 (0.02) | 0.1 (0.3) | 0.5 | n.d. | n.d. |
| β-gurjunene | 17334-55-3 | 1428 | 2.9 (1.2) | 7.4 (4.6) | n.d. | 0.7 (0.7) | 0.1 (0.3) | 15.4 | n.d. | 2.2 (2.8) |
| α-himachalene | 3853-83-6 | 1447 | 50.8 (11.8) | 28.8 (1) | 37.8 (8) | 40.2 (12.3) | 14.1 (23.8) | 28.8 | 0.3 (1.2) | 52 (26.7) |
| β-humulene | 116-04-01 | 1454 | 0.9 (1) | 1.4 (0.04) | n.d. | 0.3 (0.3) | n.d. | 1.1 | n.d. | 0.3 (0.5) |
| γ-gurjunene | 22567-17-85 | 1469 | 2.9 (4.9) | 2.5 (0.5) | n.d. | 11.6 (11.6) | 10.8 (18.5) | 3.2 | n.d. | 0.9 (1.2) |
| α-farnesene | 26560-14-5 | 1477 | 0.3 (0.4) | 0.4 (0.02) | n.d. | 8.3 (3.5) | 0.2 (0.7) | 0.3 | n.d. | 2.4 (2.7) |
| β-selinene | 17066-67-0 | 1478 | 0.7 (0.8) | 3.1 (0.2) | 0.7 (0.1) | n.d. | n.d. | 2.2 | n.d. | 0.3 (0.5) |

Laboratory samples were obtained for sieved soil samples from Organic (O) horizon (0–5 cm) and Topsoil (A) horizon. Field samples were obtained in the field with a dynamic chamber placed above uncover soil surface. RI stands for retention index. The numbers inside the brackets indicate the standard deviation of the % contribution obtained from each sample

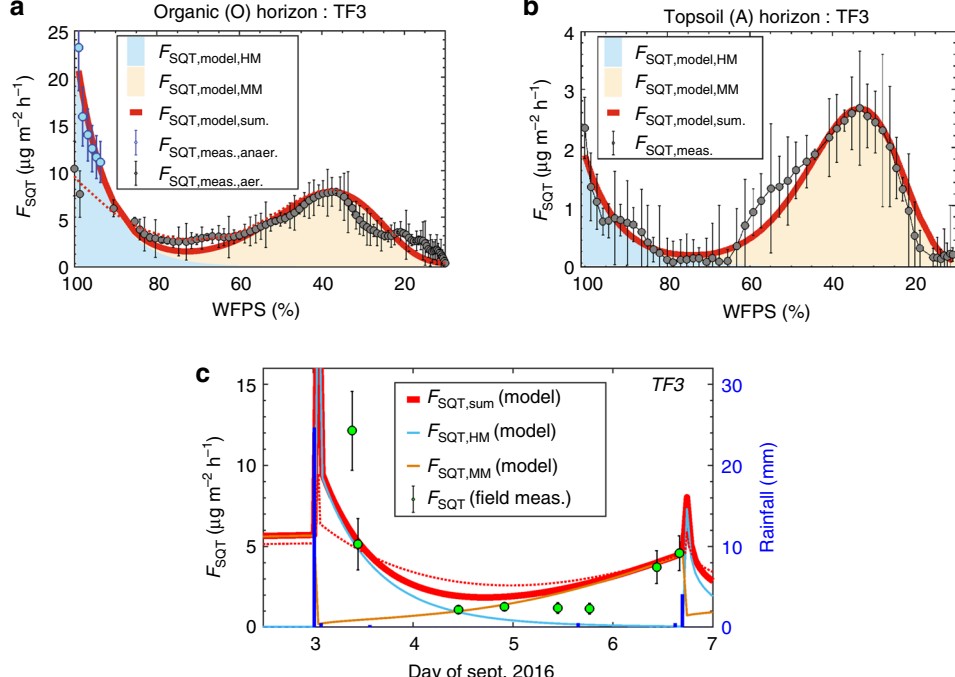

**Fig. 5** Emission model over laboratory and field conditions. **a**, **b** SQT emissions from Amazonian soils are a function of two processes: an emission burst that occurs after a strong rain event and high moisture (HM; cyan area) and continual microbial emissions that occur at optimum soil moisture conditions over the moderate moisture range (MM; beige area). For the (O) horizon (**a**, **c**), a rain event may (red line as model output) or may not (red dashed line as model output) activate anaerobic response, so both laboratory experiments (blue points for anaerobic/gray points for aerobic; error bars indicate the standard (N = 4) of the measurements) were simulated by the model. In **c**, $F_{SQT,sum}$ was calculated by assuming anaerobic activation down to 90% WFPS and that both (O) and (A) horizons (Fig. 3a, b) contribute to the release of SQT. The green bullets and bars in **c** indicate the means and standard of the field measurements

α-gurjunene, β-caryophyllene, α-humulene, α/β-cubebene, α-himachalene, and β-elemene, with their cumulative emission flux matching the emission pattern and strength of the simulated emissions. In addition to TF3, field measurements were conducted at the strong emitter soil TF1. Similar to TF3, the emission strength was predicted reasonably well by our model algorithm (model prediction: 114 μg m⁻² h⁻¹, field measurements 100 ± 56 μg m⁻² h⁻¹).

**Atmospheric model.** Since field measurements of the soil surface flux matched closely the SQT sources predicted by our new algorithm, a two-year soil SQT emission flux was compiled for a seasonal comparison with SQT emissions from the treetop canopy, as simulated by a widely used code (MEGAN v2.04) within a global model (EMAC) (Fig. 6). The modeled data from both soil and canopy encompassed two wet seasons (February–June) and two dry seasons (July–December) the last of

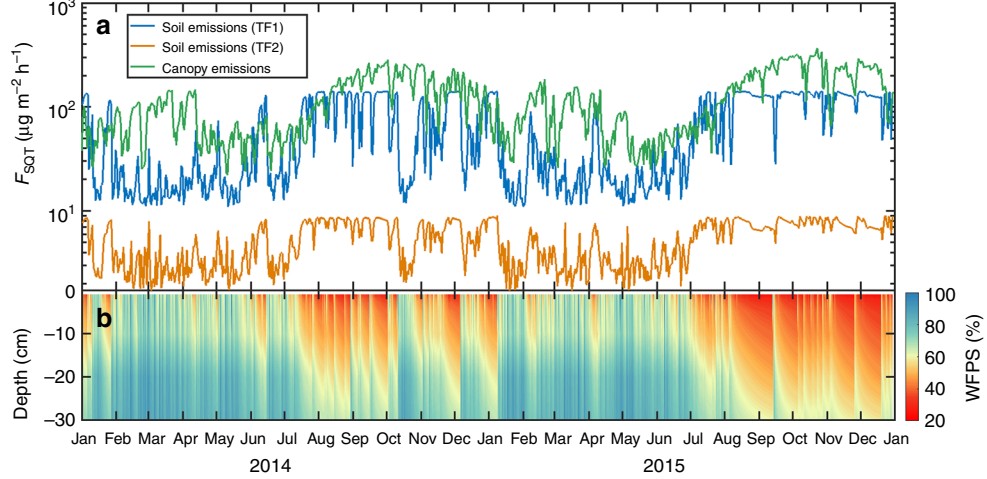

**Fig. 6** Daily averages of SQT emissions from ATTO soils and treetop canopy. Emissions from TF1 (blue line; **a**) and TF3 (orange line; **a**) were simulated with the use of WFPS at the ATTO site (**b**). Canopy emissions (green line) were calculated with the MEGAN model. WFPS is a result of hydrological modeling over the volumetric water content data at the ATTO site

which (2015) was impacted by El Niño conditions. Thus, seasonal and climatic variation of both sources could be examined. Both soil and canopy emissions reach a maximum during the dry season. Remarkably, soil emissions from TF1 are closely comparable to the tree canopy emissions, even exceeding them during the transition from wet to dry season.

**Forest canopy model**. The average dry season SQT emissions from TF soils (44.9 $\mu g\,m^{-2}\,h^{-1}$ considering emissions from TF1, TF2, TF3, and TF4) were incorporated into a simple forest canopy column model[34] (FORCAsT (forest canopy atmosphere transfer)) in order to evaluate the significance of soil SQTs to air chemistry. The majority of these highly reactive species were lost through ozonolysis within the canopy space, with a maximum of 1.5% of soil emissions escaping the canopy just before dawn when photochemistry commences and around 0.2% through most of the day as reaction rates reach a maximum. Soil emissions of SQTs account for as much as 50% of $O_3$ reactivity at the soil surface and a relatively constant 30% during daylight hours. Overnight, soil SQTs contribute nearly 40% of $O_3$ reactivity at the top of the canopy, but this is substantially reduced to 0.5–1% during daylight hours as the SQTs are rapidly consumed within the canopy. The high reactivity of SQTs results in substantial enhancements in the concentrations of condensable reaction products that will partition into the aerosol phase and potentially act as cloud condensation nuclei. SQT-derived oxidation products increase the total concentration of condensable species at the top of the canopy by 30–40% at night when reaction rates are low for other emitted compounds decreasing to around 20% at dusk when the contribution from other species is highest.

## Discussion
Strong SQT emissions from Amazonian soils have been quantified from laboratory investigations. A reproducible emission pattern as a function of WFPS was recognized and an empirical emission algorithm was created assuming different microbial processes under two different moisture regimes (high moisture/initial burst HM and medium moisture/drying out MM).

The algorithm was developed based on desiccation experiments from sieved soils. While this approach allows the quantification of purely soil-emitted VOCs, plant–microbe interactions, and the possible role of an intact web on soil emissions is not addressed

here. Other soil organisms (such as fungi, roots, micro, and macro fauna) may influence the net effect of SQT emissions from soils in neotropical forests and it remains to be tested how the algorithm, evaluated with one in situ measurement period, will perform under the seasonal changes of soil composition, microbiome community, and plant developmental stages. Nonetheless, the emission pattern and strength forecasted at Fig. 5c indicates that laboratory incubations can be used as proxy for emissions in the field.

Our samples were fumigated with VOC mixing ratios at levels that have been previously measured in the field, although due to experimental limitations, SQTs were not introduced in the headspace of the chambers. Assuming a compensation point between ingoing air and within soil SQT concentrations, the creation of atypical concentration gradients could possibly increase the net emission rate. Apart from the physical processes, biological influences may add to the uncertainties. Shown in Fig. 4, the samples S13 and S14 of TF5 (also S16 and S17 of TF6) were analyzed with different experimental setups (Exp.Set.1 and Exp.Set.2, respectively (see Methods for details)). The emission rates display the same behavior, and the emission rate differences, usually within the error bars, can be primarily attributed to inter-sample variability. Another possible reason could be the absence of VOCs in the main ingoing airstream of S13. In Exp.Set.2 we did not fumigate and therefore the lower production of SQTs observed (compared to S14; Exp.Set.1) could potentially indicate a connection between ambient-other than SQT-VOCs and microbial production in both regimes.

The initial emission burst of SQTs observed at the beginning of each experiment (HM regime) has been variously ascribed[35] to hypo-osmotic stress response of the soil microbes, the replacement of headspace air by water and its release back to the atmosphere and activation of rapid, intermediate, and/or delayed responding soil microbes[36]. Chemical speciation of SQT emission rates obtained in the field during September 2016 (TF3) showed distinctly different SQT species were emitted 6 h after a strong rainfall event compared to the following dry days, showing that different (and possibly multiple) processes are responsible for the SQT emission bursts following rainfall. Fungal emissions are exceptionally strong during their initial growing phase[30] and may reflect the highest SQT species diversity and strongest discrepancy between simulated and measured emissions that has

been observed directly after the rainfall. The SQT emissions may therefore be even stronger in seasons and regions with rapid fungal growth rates.

Under the MM regime, SQTs displayed optimum emissions at certain WFPS reproducibly, thus indicating the most favorable conditions of microbially produced SQTs. The monitoring of 16S- and 18S rRNA transcripts was undertaken as a practical means to study microbial community responses to environmental variables[35,37]. While some taxa do not demonstrate a linear correlation between rRNA abundance and microbial activity[38], ribosomes indicating potential cell activity in a community are more abundant than in dormant cells[39]. At the community level, an increase in rRNA content is assumed to reflect greater protein production over time, and therefore a proxy for activity. Our experiments demonstrated a variation of transcript abundance between two ecologically diverse soils, and in addition, they showed a peak of transcription coincident with SQT peak emissions in the MM regime. This strongly suggests that microbial activity inferred from ribosomal RNA can be associated with SQT production and release.

Emission rates from two TF soils, simulated over the course of 2 years, indicated stronger emissions during the dry season when prevailing conditions led to the WFPS associated with optimum emissions. During the El Niño year of 2015, extended drought conditions led to a slight decrease of SQT emission rates, as the field WFPS dropped to about 25–30% for the organic (O) horizon. Nonetheless, this falls within the uncertainty limits for the optimum WFPS observed in the laboratory and the direct implications of extended drought to microbial activity and subsequent SQT release requires further investigation. The emission strength, however, was validated with field measurements for both TF3 (ATTO site) and the strongest emitter TF1. The comparison with an established emission algorithm for vegetation (MEGAN) indicated that soil SQT emissions from TF1 are of comparable magnitude and so could rival the canopy emissions at the same location under certain conditions. The inherent uncertainties included in this comparison primarily originate from the temperature dependency (β-factor) that is used for vegetation emissions, and the large variability that has been observed for the soil emissions. To the best of our knowledge, no studies have directly addressed and determined the temperature dependency of SQT emissions from Amazonian vegetation. Hence, the present implicit assumption of a common and stable season independent β-factor = 0.17 (as used in MEGAN for all SQT species) can be considered a first approximation with large-associated uncertainty as it has been shown that the β-factor can vary significantly day to day, and with atmospheric conditions[9]. In addition to canopy model uncertainties, soil emissions displayed significant variation in strength between the sites investigated, with emission estimates ranging from a few $\mu g\, m^{-2}\, h^{-1}$ to orders of magnitude higher. In general, soil emissions seem to be primarily location and microbial activity specific, as demonstrated by TF4 and TF5.

The average dry season SQT emissions from Terra Firme soils were incorporated into a simple forest canopy column model[33]. The results indicated that soil SQTs dominate the ozone reactivity close to the forest floor. Soil emissions from TF1 alone could account as much as 75% of $O_3$ reactivity (soon after dawn) close to the soil surface and have a relatively constant contribution of 50% during daytime. While a small fraction of these emissions can escape the canopy, their oxidation products will be considerable at the canopy top. The model thus suggests that not only do emissions of SQTs from Terra Firme soils substantially affect atmospheric chemistry within the canopy, but that the effects have the potential to alter regional chemistry, clouds, and hence climate.

The implications of strong soil–atmosphere SQT fluxes from tropical forest soils are considerable. Ecosystem emission fluxes of SQTs and their reaction products to the atmosphere above the forest could be much larger than the currently considered canopy source, which could contribute a part of the large reported missing OH reactivity[40]. Beneath the canopy, the soil SQTs react rapidly with downward-mixed ozone, impacting the oxidative capacity within the forest[41]. Chemical reaction with soil SQT emissions could dominate $O_3$ reactivity. Furthermore, any organic particles that are formed via SQT-ozone reactions have the potential to grow to become effective cloud condensation nuclei[5,42]. Hence a connection between soil microbial activity, rainfall, SQT emissions, and clouds is postulated. We consider SQT emissions at the soil–atmosphere interface to be an important unaccounted biogeochemical process that connects microbial emissions to atmospheric chemistry and physics.

## Methods

**Soil characterization**. Soil sampling followed a standard protocol[21]. The soil types were characterized by using the world reference bases for soil resources[43]. Soil samples were collected from pits of 2 m depth, dug at the vicinity of the ATTO site (for specific locations, see Supplementary Table 1). Exchangeable cations and sum of base cations were determined using the silver-thiourea method[44] (cmol kg$^{-1}$), with elemental concentrations measured by atomic absorption spectroscopy (AAS) calibrated using suitable reference materials and blanks. For each soil, weathering conditions were determined using a total reserve bases weathering index ($\Sigma_{RB}$). This index takes into account the total cation concentration in the soil extractable by strong acid ($H_2SO_4$) and $H_2O_2$ digestion and adding this to the reservoir of exchangeable cations. Soil clay content was determined using the pipette method[45]. The quality index $\Pi$[21] is a semi-quantitative soil physical quality index that adds up scores of four soil physical properties (effective depth, soil structure, topography, and anoxia), which influence soil morphological characteristics related to pedogenesis. Higher values indicate less aerated soil conditions and more frequent anoxia.

**Experimental setup**. A dynamic laboratory incubation system allowing automated measurement of 38 samples was used to investigate the release and uptake rates of VOCs from soil. In addition, four samples were measured with a slightly different setup that is described below (see Supplementary Table 1 for an overview of all 42 samples). The chamber system that was used for the majority of the samples is described in detail elsewhere[22]. Briefly, this flow-through system allows control of soil temperature and atmospheric conditions (composition (e.g., VOC mixing ratio, $O_2$ content, and relative humidity)), and allows calculation of moisture content by tracking the evaporation flux (the difference in relative humidity between inlet and outlet air) from soil[46,47]. In a standardized incubation procedure, 80 g of sieved (2 mm mesh) field moisture and root-free soil was wetted to saturation and temperatures were held constant at either 20 °C or 30 °C under VOC-free air (i.e., normal atmospheric $O_2$ concentration) and nitrogen-pressurized gas standard (i.e., $O_2$ and VOC-free air) (6.0, Westfahlen AG, Germany). The presence of roots in the samples analyzed may induce unnatural emissions that originate from the rhizosphere and would lead to difficulties in separating soil and root emissions. Therefore, all fine roots were carefully removed from the samples, prior to their analysis. Potential remaining fine roots may increase the PTR-MS signal but their potential abundance (<0.1%) is not expected to induce the release of VOCs in a substantial manner.

The VOC-free air was produced from a zero air generator (PAG 003, Ecophysics, Switzerland), which is free of particles, and low in water (−30 °C dew point), $NO_x$, $SO_2$, $O_3$, and CO. Six soil incubation chambers (Teflon foil coated, diameter = 9.2 cm, height = 13.75 cm) were used for each experiment while one reference chamber without soil was used to monitor the background mixing ratio of the VOCs. Because of the low-water content of incoming air flushing the incubation chambers, soils lost water over time at a rate that could be tracked by measuring the difference in relative humidity between incoming and outgoing air streams and multiplying by the rate of air flow kept constant by mass flow controllers; see detailed description below. Gravimetric water loss was converted to percent water-filled pore space using field collected bulk density data and the mass of water required to reach field capacity. While the laboratory soil density can be slightly different after sieving, the field bulk density was chosen as characteristic for each soil type to be consistent with previous studies[24]. The directly measurable quantity of field bulk density was therefore used to derive gravimetric soil moisture in this work. It normally took 4–7 days to dry the soil completely, allowing us to investigate fluxes over the drying out period. While such conditions may not represent the natural diffusion-driven exchange, it has been shown that our flow-through experimental setup can reliably quantify gaseous emissions from soil, such as NO[22,24].

For simultaneous quantification of gas uptake and production rates, we used a VOC calibration gas standard (14 components; Apel-Riemer Environmental, USA, diluted to the 0.1–1 ppb range) diluted with the zero air (or $N_2$) flow. Depending on the experiment, low and atmospherically relevant VOC mixing ratios (0.1–1 ppb) were introduced to the main airstream of VOC-free air. In addition to VOCs, $CO_2$ was constantly introduced in the main airstream. The soils were fumigated with field-relevant (measured) $CO_2$ mixing ratios (0–5 cm: 400–3000 ppm, 10–50 cm: 1000–14,000 ppm) during the drying out process. A custom-made electronic control system (V25, Max Planck Institute for Chemistry) was used to regulate the introduced mixing ratios for every measuring cycle. The air from the outflow of each chamber was monitored by the PTR-MS. Each chamber was monitored for 10–15 min before switching to the next chamber.

The airstream flows were regulated by mass flow controllers (Bronkhorst, Wagner Mess- und Regeltechnik GmbH, Germany), which were calibrated by a primary air flow calibrator (Gillan Gilibrator 2, Sensidyne, USA). A total of 5 l/min flow was split into two streams. During sampling, 2.5 l/min were directed through the measuring chamber. The chambers that were not actively being sampled were continuously flushed with 0.5 l/min in order to continue the drying process (though at slower rates). The outlet of all chambers was connected to a single line to which the proton transfer reaction–mass spectrometer (PTR-MS) and an ultraportable greenhouse gas analyzer (Los Gattos Research Inc., USA) that measured water vapor and $CO_2$ were connected.

Experiments S3, S4, S13, and S16 were conducted with a different experimental setup (hereafter called EXPSET2) that uses the same operating principle. The same standardized procedure and soil handling was followed. The difference with the previously described system is that instead of zero air generator, synthetic air (6.0, Westfahlen AG, Germany) was used as zero air and the chambers were made from 100% Teflon (inner diameter = 10 cm; chamber volume = 500 ml). The same chambers were used for the determination of the emission rates in the field (see "Flux measurements" in the field below). In total, three main air streams (each 1 l/ m of VOC-free air) were split into two individual streams each of 0.5 l/min. In total, six chambers were operated. For these experiments, each main airstream was having a separate zero chamber that was used for the determination of background signal. The outlet of all chambers was connected to a single line to which the proton transfer reaction–mass spectrometer (PTR-MS) was connected.

**PTR-MS measurements**. Mixing ratios of VOCs were quantified on-line using a high-sensitivity PTR-MS (IONICON Analytik GmbH, Austria). Molecules (R) with higher proton affinity than water (691 kJ/mol) were ionized inside a low-pressure (2.2 mbar) drift tube with hydronium ions ($H_3O^+$) produced in the ion source via electrical discharge of water vapor. The electrical field in the drift tube accelerates the ionized molecules that are finally detected by a quadrupole mass spectrometer at their protonated molecular mass ($RH^+$). A detailed description of the operating principle can be found elsewhere[48].

The PTR-MS was operated under standard conditions ($E/N = 117$ Td, 600 V, 2–2.2 mbar). Humidity-dependent calibrations were performed by use of a calibration gas standard (Apel and Riemer Environmental, USA) containing methanol, acetonitrile, acetaldehyde, acetone, dimethyl sulfide, isoprene, methyl vinyl ketone, methyl ethyl ketone, benzene, toluene, o-xylene, and a-pinene. The mixing ratios of the molecules that were not present in the calibration gas (e.g., SQT) were calculated with the use of an experimentally derived transmission curve (i.e., while calibrations were used for the species in the calibration standard, SQTs were calculated by applying the instrument's transmission curve which is a commonly used procedure for species without stable calibration sources. The transmission curve takes into account the changes in ion transmission efficiency through the detector which change with size.). The background signal was determined with VOC-free air, generated from a catalytic converter (Platinum pellets, 400 °C).

SQT ions were detected at $m/z$ 205. The reaction of SQT with $H_3O^+$ under typical drift tube conditions results in multiple fragments due to both dissociative and non-dissociative proton transfer. Therefore, detection efficiency is lower than more robust VOCs. Despite the high correlation ($R^2 > 0.9$) with the major fragment ion ($m/z$ 149), only the parent ion ($m/z$ 205) was used for the calculation of the mixing ratios. The relative abundance was derived from literature (30%)[9,49], together with the reaction rate constant ($k_{SQT+H3O+} = 3 \times 10^{-9}$ molecule $cm^3 s^{-1}$)[9,50,51]. The fragmentation pattern of SQT inside the drift tube depends on both E/N ratio and the structure of the molecule[52]. The fragmentation of the dominant SQT species newly identified in this study (i.e., α-gurjunene, α-himacalene, and β-eudesmene) has not previously been characterized. Hence, larger uncertainties are expected in the final calculation of SQT mixing ratios (≈50%).

Experiments S3, S4, S13, S14, S15, and S16 were conducted with a different but same model quadrupole PTR-MS system (hereafter referred as PTR-MS2; IONICON Analytik GmbH, Austria) that was tuned for the same drift tube conditions ($E/N = 117$ Td, 600 V, 2–2.2 mbar). The transmission efficiency for PTR-MS2 was lower for sesquiterpenes ($T_{SQT,PTR-MS1} = 0.3$, $T_{SQT,PTRMS2} = 0.19$) and therefore decreased precision ($Pres._{PTR-MS1} = 9.4 \pm 3.1\%$; $Pres._{PTR-MS2} = 15.9 \pm 4.4\%$) was observed.

Shown in Fig. 4, the samples S13, S16, and S14, S15 were analyzed with EXPSET1 and EXPSET2, respectively (using PTR-MS2). The emission rates display the same behavior and the emission rate differences (usually within the error bars) can be primarily attributed to inter-sample variability. Another possible reason

could be absence of VOCs in the main airstream of S13. In EXPSET2, we did not fumigate with environmentally relevant VOCs, therefore the lower production of SQTs observed (compared to S14; EXPSET1) could potentially indicate a connection between ambient VOCs and microbial production and future studies shall focus in identification of possible microbial mechanisms and pathways.

**GC-MS measurements**. Since the PTR-MS is unable to separate SQTs into individual species, additional GC-MS analysis was vital for the identification of the individual SQT structures and direct comparison of the measured emission rates (Supplementary Fig. 1). Adsorbent tube samples filled with Quartz wool/Tenax TA/Carbograph 5TD (Markes Environmental) were used to collect air samples in parallel to the on-line measurements and these were subsequently analyzed off-line. The adsorbent tube samples were analyzed using a thermal desorption instrument (Perkin-Elmer TurboMatrix 650, Waltham, USA) attached to a gas-chromatograph (Perkin-Elmer Clarus 600, Waltham, USA) with DB-5MS (60 m, 0.25 mm, 1 μm) column and a mass selective detector (Perkin-Elmer Clarus 600T, Waltham, USA). The sample tubes were desorbed at 300 °C for 5 min, cryofocused in a Tenax TA cold trap (−30 °C) prior to injecting the sample into the column by rapidly heating the cold trap (40 °C $min^{-1}$) to 300 °C. A five-point calibration was performed using liquid standards in methanol solutions. Standard solutions (5 μl) were injected onto adsorbent tubes and then flushed with helium (80–100 ml $min^{-1}$) for 10 min to remove the methanol. The following SQTs were included in the calibration solutions: longicyclene, iso-longifolene, α-gurgunene, β-caryophyllene, aromadendrene, and α-humulene. Unknown sesquiterpenes were tentatively identified based on the comparison of the mass spectra and retention indexes (RIs) with NIST mass spectra library (NIST/EPA/NIH Mass Spectral Library, version 2.0). RIs were calculated for all SQTs using RIs of known SQTs and monoterpenes as reference. These tentatively identified SQTs were quantified using response factors of calibrated SQTs having the closest mass spectra resemblance.

**Calculation of release and uptake rates**. The release and uptake rates of the investigated VOCs were calculated using the following equation:

$$F_{VOC} = Q \cdot \frac{(C_{out} - C_{in})}{A} \tag{1}$$

Where Q is the gas flow rate through the measured chamber (in $m^3 h^{-1}$), $C_{out}$ and $C_{in}$ are the VOC concentrations in the air exiting chambers holding soil samples (soil) and empty chambers without soil (reference chamber), respectively (in μg m$^{-3}$) and A is the headspace area of each chamber ($m^2$). The final release and uptake rates were calculated in μg m$^{-2}$ h$^{-1}$. $C_{out}$ and $C_{in}$ were calculated from the average of the last four data points before the chamber switch.

**Calculation of water-filled pore space**. The mass of soil was determined gravimetrically at the beginning ($t_0$) and end ($t_s$) of the experiment as $m_{soil}$ ($t_0$) and $m_{soil}$ ($t_s$). Over the course of the drying out for each experiment, the shape of the $H_2O$ signal over incubation time was converted into mass of wet soil by the use of the $H_2O$ vapor mass balance of the dynamic chamber which was further developed as a recursion formula[22] as shown below:

$$m_{soil}(t_i) = m_{soil}(t_0) + g \cdot \Big( s_{H2O,cham}(t_i) \cdot (V + Q \cdot \tfrac{t_i - t_{i-1}}{2})$$
$$- s_{H2O,cham}(t_{i-1}) \cdot (V + Q \cdot \tfrac{t_i - t_{i-1}}{2}) \tag{2}$$
$$- Q \cdot \tfrac{t_i - t_{i-1}}{2} \cdot (s_{H2O,ref}(t_i) + s_{H2O,ref}(t_{i-1})) \Big)$$

where V is the volume of the headspace of the chamber, Q the flow rate $t_i$ and $t_{i-1}$ the incubation time in seconds, and $s_{H2O,cham}(t_i)$, $s_{H2O,cham}(t_{i-1})$, $s_{H2O,ref}(t_i)$, and $s_{H2O,ref}(t_{i-1})$ is the $H_2O$ signal in the soil chamber and reference chamber measured by the ultraportable greenhouse gas analyzer (Los Gattos Research, USA) at $t_i$ and $t_{i-1}$, respectively. The factor g was calculated as:

$$g = \frac{m_{soil}(t_s) - m_{soil}(t_0)}{V \cdot \big[ s_{H2O,cham}(t_s) - s_{H2O,cham}(t_0) \big] + S_0} \tag{3}$$

where $s_{H2O,cham}(t_s)$ and $s_{H2O,cham}(t_0)$ is the signal of $H_2O$ in the soil chamber at $t_s$ and $t_0$, respectively. And $S_0$ was calculated as:

$$S_0 = Q \cdot \left[ \sum_{i=1}^{i=n} (\tfrac{t_i - t_{i-1}}{2} + \tfrac{t_{i-1} - t_{i-2}}{2}) \cdot \big( s_{H2O,cham}(t_{i-1}) - s_{H2O,ref}(t_{i-1}) \big) \right] \tag{4}$$

The mass of soil was converted into water-filled pore space [%], $WFPS_{lab}$ by

$$WFPS_{lab}(t_i) = \frac{m_{soil}(t_i) - m_{soil}(t_s)}{m_{soil}(t_s)} \cdot \frac{100}{\theta_s} \tag{5}$$

where $\theta_s$ is the saturated gravimetric water content in the laboratory at the beginning of the experiment and $m_{soil}(t_s)$ equals the mass of soil at the end of the experiment, respectively. $\theta_s$ was determined experimentally for each homogenized

soil sample (sieved through a 2 mm mesh) followed by the addition of $H_2O$ until the surface of particles was covered by a tiny film of water.

**Emission model.** For a given temperature, our algorithm for modeling SQT fluxes incorporates the emission equation as a function of WFPS that has been developed previously for NO[24], with the addition of a term that describes the exponential decay of the emission burst upon wetting.

$$F_{SQT}(WFPS) = aWFPS^b \exp(-cWFPS) + d\exp(-fWFPS) \qquad (6)$$

The parameters $a$, $b$, $c$, $d$ and $f$ were related to the observed values:

$$a = \frac{F_{SQT}(WFPS_{opt})}{WFPS_{opt}^b \exp(-b)} \qquad (7)$$

$$b = \frac{\ln\left(\frac{F_{SQT}(WFPS_{opt})}{F_{SQT}(WFPS_{upp})}\right)}{\ln\left(\frac{WFPS_{opt}}{WFPS_{upp}}\right) + \frac{WFPS_{upp}}{WFPS_{opt}} - 1} \qquad (8)$$

$$c = \frac{-b}{WFPS_{opt}} \qquad (9)$$

Here, $F_{SQT}$ is the moisture-dependent emission, $F(WFPS_{opt})$ is the highest emission which is observed at $WFPS_{opt}$ and $F(WFPS_{upp})$ is the emission at half maximum, when $WFPS_{upp} > WFPS_{opt}$. The constants $d$ and $f$ were empirically derived by an exponential fit over the first 6 h after the initial wetting.

**Flux measurements in the field.** Flux measurements at TF3 (ATTO is the Amazonian Tall Tower Observatory, located about 150 km northeast of Manaus, Brazil; see http://www.mpic.de/en/research/collaborative-projects/atto.html) and at TF1 were quantified with the use of custom-made, non-transparent Teflon chambers (inner diameter = 10 cm; chamber volume = 500 ml; same chambers as EXPSET2) and application of Eq. (1). On the top of each chamber were two ports that were used for the ingoing and outgoing airstream. The chambers were placed directly over litter free soil and synthetic air was pumped through the chamber with a rate of 792 ± 164 cm$^3$ min$^{-1}$. To avoid root damages and hence artificial emissions, the chambers were not installed inside a collar, but Teflon foil was used to close the surrounding of the chamber and its connection to the soil surface to achieve the minimum disturbance of the soil bellow the chamber. At the ATTO site, synthetic air from pressurized gas bottle was used for the ingoing air. Applying an active flux though the chamber may result in an overestimation of the quantified VOCs but it will not affect the emission pattern of sampled species. Due to the remote location of some of TF1, Teflon bags were filled with zero air prior to each experiment and connected to the inlet port before sampling. The regulation of the synthetic air inflow was made with a calibrated rotameter. A T-piece at the outlet of the chamber was used to ensure an overflow while an adsorbent tube (filled with Quartz wool/Tenax TA/Carbograph 5TD; Markes Environmental) was sampled. A total volume of 2.5–3.5 l was collected at a sampling flow rate of 167 ± 8 cm$^3$ min$^{-1}$. To account for flux variability due to possible soil heterogeneities, three separate samples were simultaneously (0.5–1 h difference) taken from areas in the vicinity of the pit. We note that the experiments were set up directly after the rain event (Fig. 5c) and hence the soil was not covered during when rainfall occurred.

Assuming a compensation point between ambient air and within soil SQT concentrations, the use of zero air used for the ingoing air, could potentially lead to an overestimation of the fluxes measured. Nonetheless, the emission pattern forecasted at Fig. 5c indicates that despite field experimental restrictions, laboratory incubations can be used as proxy for emissions in the field.

Upon collection, the adsorbent tubes were shipped to Finland and analyzed between 3–4 weeks later with the aforementioned method. Due to the discovery of the high abundance of α-gurjunene in the laboratory experiments, the GC-MS quantification method for the field measurements was performed with an authentic liquid standard.

The error propagation for the presented points has been calculated as follows:

$$ErF_{SQT} = \sqrt{ErQ^2 + ErGC^2 + Std^2} \qquad (10)$$

where $ErF_{SQT}$ is the total error for each point, $ErQ$ is the uncertainty over the sampling flows, $ErGC$ is the uncertainty due to quantification of SQT, and Std is the standard deviation over the triplicates sampled at each given time. In Fig. 5c all data points are the result three samples, apart from the points presented for the 3rd of September, which are the product of duplicates.

**Hydrological model.** One-dimensional variably saturated water flow into soils is described by the 1D-Richards equation[53]:

$$\frac{\partial \theta}{\partial t} = \frac{\partial}{\partial z}\left[K(h)\left(\frac{\partial h}{\partial z} + 1\right)\right] \qquad (11)$$

where $\theta[L^3 L^{-3}]$ is the water content, $t[T]$ is time, $z[L]$ is the vertical spatial coordinate, positive upwards, $h[L]$ is the pressure head, and $K(h)[LT^{-1}]$ is the unsaturated hydraulic conductivity function.

The numerical solution of Eq. (2) requires the definition of the soil hydraulic properties (SHPs), i.e., the water retention curve $\theta(h)$ (SWRC) and the unsaturated hydraulic conductivity curve $K(h)$ (HCC). We used the van Genuchten–Mualem (VGM) model[54] for all simulations. The SWRC and HCC are given by the equations:

$$\theta(h) = \begin{cases} \theta_r + (\theta_s - \theta_r) \cdot (1 + |\alpha h|^n)^{-m}, & h < 0 \\ \theta_s, & h \geq 0 \end{cases} \qquad (12)$$

$$S_e = \frac{\theta(h) - \theta_r}{\theta_s - \theta_r} \qquad (13)$$

$$K(S_e) = K_s \cdot S_e^l \cdot \left[1 - \left(1 - S_e^{\frac{1}{m}}\right)^m\right]^2 \qquad (14)$$

where $\theta_s$ and $\theta_r[L^3 L^{-3}]$ are saturated and residual water contents, respectively, $\alpha[L^{-1}]$, $n[-]$, $m[-]$ and $l[-]$ are shape parameters, $m = 1 - \frac{1}{n}, n > 1$, and $S_e$ [-] is effective saturation.

The Richards equation was solved numerically by using the Hydrus 1D software[55]. The 1 m soil profile was divided in three different layers: upper layer 0–10 cm, middle layer 10–20 cm, and bottom layer 20–100 cm. Each layer is described by a unique set of SHPs. A free-drainage boundary condition was used for the lower boundary (1 m) and an atmospheric boundary condition was used at the soil surface. Measured rainfall and evaporation were used as specified fluxes across the soil surface. The initial condition was specified as pressure head distribution given by preliminary simulations (warm up period) in order to reflect realistic conditions.

The model was calibrated against field measured water content values at 10, 20, and 100 cm depth. The objective function to be minimized for determining the vector of unknown parameters $\vec{p}$ (the SHPs for the three layers) is given by the weighted-least-squares formulation

$$O(\mathbf{p}) = \sum_{i=1}^{N} w_i r_i(\mathbf{p})^2 \qquad (15)$$

where $N$ is the number of data points in the objective function, $r_i$ are the residuals, i.e., the differences between the observed and the model-predicted data, and $w_i$ are weights which reflect the reliability of the individual measurements. Iterative minimization of Eq. (15) with respect to the parameter vector $\vec{p}$ was achieved with the SCE–UA global search scheme[56]. Water content and WFPS at each depth was calculated by conducting forward simulations with the calibrated model.

**Global model of treetop emissions.** The EMAC (ECHAM5/MESSy Atmospheric Chemistry) model has been used to estimate plant emissions of SQTs in the location of the ATTO tower during the years 2014 and 2015. The EMAC model is based on the 5th generation European Center Hamburg general circulation model (ECHAM5[57]) and the Modular Earth Submodel System (MESSy[58]). In the present study, we applied EMAC (ECHAM5 version 5.3.02, MESSy version 2.52) in the T106L31-resolution, i.e., with a spherical truncation of T106, corresponding to a quadratic Gaussian grid of ~1.1 by 1.1 degrees in latitude and longitude, with 31 vertical hybrid pressure levels up to 10 hPa. The model dynamics have been weakly nudged toward ERA-Interim data[59]. In this study, the model was run without photochemical calculations, thus merely as general circulation model (GCM) to represent emission fluxes. In addition, the submodel SCOUT (Stationary Column OUTput) to extract data at the location of the ATTO tower, and the MEGAN submodel were applied. The latter submodel is the implementation of the MEGAN model (Model of Emissions of Gases and Aerosols from Nature, version 2.04[60]) into EMAC, where input from the GCM is used to estimate biogenic emissions of tracers[61,62]. The model was run for 2 years (2014–2015) and the emissions at the location of the ATTO tower estimated by MEGAN were outputted at 1-hourly frequency via the SCOUT submodel. The SQT emissions were integrated to estimate the total source calculated by MEGAN in this location.

**Molecular analysis.** Subsamples of TF4 and TF5 soil were collected from the chambers during the desiccation experiment. Soil was sampled at moments chosen due to sesquiterpene emission profile: after wetting, after 18 h and 48 h approximately, representing $ESQT_{max}$ and $ESQT_{min}$ values. Subsamples were collected from five randomly chosen points within the chamber providing a total of 1.5 (±0.25) g dry weight of soil, collected in 15-ml Falcon tubes and immediately snap-frozen in liquid $N_2$ and stored at −80 °C before molecular analysis. Sampling

scheme was designed to provide composite pseudo-replicate samples representing the community present in the chamber at each moment. Half of the soil in each sample was used to calculate gravimetric moisture and the other half for RNA extraction. RNA was extracted with a total RNA Isolation Kit (RNA PowerSoil®, MO BIO Laboratories Inc., USA). Qubit 3.0 fluorometer® (Invitrogen, USA) was used to assess RNA quantity with respective assay HS kits. An aliquot of RNA was reverse transcribed to cDNA using SuperScript® VILO™ Master Mix (Invitrogen, Karlsruhe, Germany) after DNAse treatment (DNase Max, MoBIO, CA, USA). Quantification of bacterial and fungal 16S and 18S transcript abundances per gram dry soil was performed in a StepOnePlus™ real-time PCR System (Applied Biosystems, USA). Standard curves were obtained using tenfold serial dilutions from calculated 1011 copies $\mu l^{-1}$ and 1010 copies $\mu l^{-1}$ of 16S rRNA and ITS genes, respectively, obtained from Escherichia coli (DSM 30083 strain) and Sacharomyces cerevisiae (DSM 70449 strain) genomes. We applied the primers 338F/534R for 16S rRNA and FR1/FF390 for 18S rRNA, which have been used in previous studies[63,64]. Each reaction had a final volume of 20 μl containing 1× Power SYBR Green PCR MasterMix (Invitrogen, Karlsruhe, Germany), 0.2 μM of each primer, and 2 μl of cDNA twofold diluted. Bacterial and fungal transcripts were amplified according to the cycling conditions: 15 min at 95 °C, followed by 40 cycles of 30 s at 94 °C, 30 s at 53 °C (50 °C for the fungal primers) and 30 s at 72 °C (1 min of extension for the fungal primers). SYBR green fluorescence was measured after the elongation step for each run. The $R^2$ and the efficiency of amplification were 0.998 and 89.9 and 0.994 and 86.2 for bacterial and fungal standard curves, respectively. Each gene was assayed for all samples in two separate runs with duplicates for each sample, with final four replicates standardized between runs.

**Canopy model**. The FORCAsT (FORest Canopy Atmosphere Transfer) model[34] has been used to simulate soil emissions and processing of SQTs within the rainforest canopy. The model was run over 48 h for an average meteorology for September 2014, derived from in situ observations at the ATTO site, and the output of the first day discarded as spin-up. Initial concentrations of biogenic VOCs, ozone, $NO_x$, CO, and $CO_2$ within and just above the canopy were taken from observations made during this and previous measurement campaigns at this site[27,65,66]. The average height of the canopy was taken to be 40 m and it was assumed that the forest around the site was homogeneous. Soil characteristics were taken from Andreae et al.[27] and vertical distribution of leaf area was based on Kuhn et al.[67].

SQTs were introduced into the model at the soil–atmosphere interface at a constant rate of 44.9 μg $m^{-2}$ $h^{-1}$, representing the average emission flux measured from Terra Firme soils in the dry season. 10% of the emitted SQTs were assumed to be β-caryophyllene and the remainder lumped as "other SQTs". β-caryophyllene chemistry follows the master chemical mechanism for the first two generations of oxidation products with subsequent products lumped; only the initial reactions of "other SQTs" (with $O_3$, and the OH and $NO_3$ radicals) are explicitly included. FORCAsT does not include an aerosol phase but does explicitly calculate the rate of formation of low volatility oxidation products, which are assumed to condense into particles[34].

**Data availability**. The data sets within the article and Supplementary Information of the current study are available from the authors on request.

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

## Acknowledgements

We thank the Max Planck Society and the Instituto Nacional de Pesquisas da Amazonia for continuous support of the ATTO project. We acknowledge the support by the German Federal Ministry of Education and Research (BMBF contract 01LB1001A) and the Brazilian Ministério da Ciência, Tecnologia e Inovação (MCTI/FINEP contract 01.11.01248.00) as well as the Amazon state University (UEA), FAPESP, CNPq, FAPEAM, LBA/INPA, and SDS/CEUC/RDS-Uatumã. E.B. acknowledges the support of BmBf project ATTO (01LK1602B). The work of T.B. has been funded by the Deutsche Forschungsgemeinschaft (DFG) CRC 1076 "AquaDiva". P.A. acknowledges funding from FAPESP – Fundação de Amparo à Pesquisa do Estado de São Paulo. We like to thank all the people involved in the logistical support of the ATTO project, in particular Reiner Ditz, Hermes Braga Xavier, and Dr. Niro Higuchi. Finally, we like to thank Dr. Susan Trumbore, Dr. Matthias Sörgel, and Dr. Franz Meixner for the discussions during the preparation of the manuscript.

## Author contributions

E.B. had the idea, performed the laboratory experiments, analyzed the data, and wrote the paper. T.B. performed the laboratory and field measurements and was responsible for sample collection, gravimetric soil moisture determination, and microbial analysis. A.M. Y.S. performed the laboratory experiments, H.H. analyzed the chemical speciation of SQT, E.D. used a hydrologic model to quantify moisture dynamics, E.C. conducted the microbial analysis, K.A. employed a forest canopy box model to evaluate the significance of soil SQTs in air chemistry, A.P. used a global model to derive canopy emissions, M.S. and A.A. provided the meteorological data at the ATTO site, C.A.Q. and D.L.M. analyzed the physical and chemical properties of the soils, P.A. and J.B. provided the PTR-MS instrument, J.K. and J.L. supervised the project and commented the paper, J.W. had the idea, supervised the project and wrote the paper. All authors have contributed to writing and editing the manuscript.

## Additional information

**Competing interests:** The authors declare no competing interests.

