## [Peer Review File · Nature Communications]

Reviewers' comments:

Reviewer #1 (Remarks to the Author):

General comments

This study examined the contribution of neotropical soils to the atmospheric charge of biogenic volatile organic compounds in the Amazonian basin. In well-designed laboratory experiments the influences of several environmental factors such as soil moisture, temperature and atmospheric composition on sesquiterpene emissions were investigated on a large number of soil samples subdivided in diverse horizons. Based on the results the authors developed an emission model, which was further integrated in an existing modelling framework for spatial and temporal extrapolation. In addition field measurements were conducted as well as specific

RNA analyses of soil samples were to elucidate the biological origin of soil VOC exchanges.

In addition RNA analyses of soil samples were conducted to elucidate the biological origin of soil VOC exchanges. Overall the results provide strong evidence that soil bacteria emit large amounts of sesquiterpenes representing a large, hitherto overseen source of highly reactive carbons.

The results are novel and are important to scientists working in the field of atmosphere biosphere interactions. Research on VOC exchange at the soil atmosphere interface has received increasing attention during the last decade, but comprehensive studies such as the present one are sparse.

The paper is overall well written and should make a nice contribution to Nature Communication. However, prior publication some clarifications and corrections are needed.

I have been a bit disappointed by the section Discussion, which is very brief. In fact it resembles rather a listing of conclusions than a critical examination of the study in the context of the current understanding of soil VOC exchanges. On the other hand, in other sections the description of the results and methods contain many statements of data interpretation that could be moved to the section Discussion (e.g. L66 ff; L583-586. Alternatively the current section "Discussion" is maintained as it is but renamed as "Conclusions". In any case there are several directions in which the authors could have more extended the discussion. I hope my following comments listed below will help doing so.

I wonder whether the authors performed statistical analyses to check whether emissions are significantly different between soil types and horizons, sites, environmental conditions a.s.o.(see e.g. description of results L84-92)

The authors mention several times in the ms (e.g. L426 ff) that for simultaneous quantification of gas uptake and production rates low and atmospherically relevant VOC mixing ratios were introduced to the chamber air supply during laboratory experiments. However no precisions are given about the exact VOC amounts and composition that were applied during laboratory fumigation experiments and no results are given about uptake, remission or transformations of fumigated VOC.

RNA analyses strongly suggest that soil bacteria are the main sources of SQTs in the present study. Unfortunately these analyses were made only on TF4 and TF5 soils of the dryer outer region of the Amazonian basin (Cerrado-savannah ecosystem) and not on soils of the inner tropical rain forest, on which however the authors concentrated their emission studies (cf table 1). This limitation is not explained nor discussed in the manuscript.

In the lab soils were assayed under anaerobic conditions were simulated by incubation in pure Nitrogen. I guess that under natural conditions proxy anaerobic conditions (hypoxia) in the upper soil horizons will exclusively caused by water logging (water saturated soils) after flooding and heavy rainfalls. If so results from laboratory experiments under dry anoxia conditions might not really represent common conditions in nature?

The laboratory work concentrated on soil exempted from roots. While I understand the author's choice, it represents a potential limitation since the rhizosphere with associated microbes may strongly affect VOC exchange characteristics of soils. I guess that in dense forests with fine roots are everywhere.

More secific comments and questions:

L40: I suggest replacing "aid" by "affect".

L77-78: " Acetone emissions were the strongest, with release rates up to 1.5 mg m² h⁻¹; an order of magnitude higher than acetone emissions from Amazonian vegetation²⁶."

For the sake of clearness I suggest replacing "...than acetone emissions from Amazonian vegetation²⁶" by "...than acetone net fluxes measured above an Amazonian forest²⁶".

Similarly L82: please insert “fluxes” after “canopy”.

LL84-94: The differences in SQT emissions that are described in this paragraph are not backed by results from statistical analyses.

L43: I do not understand why “Potential remaining fine roots may interfere with the PTR-MS signal”?

L113-114: “Along with the production strength, the chemical diversity of SQT emissions increased. Possibly this is just a matter of detection limit?”

L123. “represent common conditions in nature” see my comment above.

L133 What are “optimum emission functions”?

L168: I suggest replacing “will” by “may” because the interference of ozone in VOC mediated plant-insect interactions have yet been demonstrated only by a limited number of studies.

Figure 1 and caption.

Fig 1a: L198-199: “a. Normalized soil fluxes of SQTs (red) and acetone (purple) as a function of WFPS. The normalized, algorithm derived emission curve is the result...”

normalized fluxes with respect to what? What is the blue line in 1a?

LL 205-206: “The asterisk in panel e denotes that the experiments were performed under aerobic conditions.” If measurements were made under aerobic conditions why in Fig 1e anaerobic is mentioned?

L277: What is meant by “...and is between area comparison parameter”

Extended data table 2: please add standard deviations or standard errors to the data where applicable (for N>1). Further I suggest showing absolute emission rates as well (+/- Stdev), for example in parenthesis below each percentage data.

Why RTs are not given for all compounds?

Extended data figures 1 and 2:

Increase size of symbols

According which criteria the authors defined data as outliers?

Extended data figure 3: Is this figure cited in the text?

Data in Figure C are shown in a very “compressed” manner. I suggest increasing the height of this figure and showing data in a normal scale in addition to the logarithmic scale. The reliability of the identification and quantification of SQTs by PTR-MS (see also L462 ff) is crucial in the present study, because the postulated high emission quantities present the most original outcome of the work. In my view a consistent quantification of PTR-MS for SQTs requires regular calibration with gaseous SQT-standards based on and controlled by GC-MS measurements made in parallel at least once per experiment on the different soil samples. Apparently no SQT standards were used for PTR-MS calibration and results from parallel GC-MS measurements are not presented for all soil samples (cf extended data table 2).

L423-424: I do not understand how field bulk density data can be applied to sieved soil samples?

L434: what was monitored?

L467: please explain “experimentally derived transmission curve”.

L481: The brand name and model of the second PTR-MS system should be quoted.

L486: This statement contradicts what is stated in line 481 (S3, S4, S13, S15, S16 were conducted with PTR-MS2).

L587-590. The statements here are somewhat confusing. Adsorbent tubes of field measurements were analysed in Finland but not those of the laboratory experiments? Where were the laboratory samples analyzed? All were analyzed with exactly the same analytical set-up and calibrated with the same authentic standards?

Additional references for consideration:

Leff JW, Fierer N. 2008. Volatile organic compound (VOC) emissions from soil and litter samples. *Soil Biology and Biochemistry* 40: 1629-1636.

Schade GW, Custer, Thomas G. 2004. OVOC emissions from agricultural soil in northern Germany during the 2003 European heat wave. *Atmospheric Environment* 38: 6105-6114.

Asensio D, Peñuelas J, Filella I, Llusà J. 2007. On-line screening of soil VOCs exchange responses to moisture, temperature and root presence. *Plant and Soil* 291: 249-261.

Reviewer #2 (Remarks to the Author):

This manuscript addresses an important topic in current biogeochemistry - how do the forests (and soils) of Amazonia interact with the atmosphere and what are the Earth system implications of these interactions? More specifically, very considerable research effort has been expended on understanding how the Amazonian forest contributes reactive hydrocarbons and other organic compounds to the atmosphere, and what the impacts of these compounds are on regional-scale atmospheric chemistry. Reactivity measurements have indicated that there is likely an additional strong source of VOCs in the forest environment, other than trees themselves. In this manuscript it is shown that forest soils may be a very considerable source of reactive sesquiterpenes to the atmosphere, at emission rates which match those from the forest canopy. This is an important and novel finding which will be of interest to a wide readership which is worthy of publication in *Nature Communications*. However, I do have one very major comment/reservation that does require attention before the manuscript can be accepted for publication - that relating to the discussion - from line 162 onwards - see below.

Specific comments that require responses:

lines 32/33: quoting modelled flux rates to one tenth of a microgram/m²/h gives a spurious sense of precision which is completely unjustified. On what basis does a value of ~100 "rival" one of ~170? On what basis is the quoted degree of uncertainty estimated?

Fig 1: What is the replication here? Are these plots from single measurements? What does the shading represent?

Fig 2 contains more information than this reviewer can process in one diagram. I am not suggesting that the figure has to be changed, but the authors should at least consider alternative ways of displaying this information. What is the degree of replication in this figure?

Fig 3 caption puts forward an explanation for the observed behavior in terms of two processes - but this is purely speculative, and should be expressed as such. No indication of the degree of replication is given.

line 64 onwards: no indication of experimental replication is given. Where were these single experiments or was there true experimental replication?

line 96-97: is the transcript abundance expected or assumed to vary linearly with bacterial/fungal activity? A little more explanation of why the transcripts were measured and what assumptions are being made about them is needed here.

lines 162/3 and lines 166/7: There is a nasty inconsistency in the arguments made here. On the one hand, the authors are claiming that SQT emissions from the forest soils may leave the forest canopy into the atmosphere - at rates that rival those from the trees themselves. On the other hand, they then say that SQTs emitted from the soils may react below and/or within the canopy, so explaining low observed ozone levels. Well, they can't have it both ways so easily....if SQT molecules are consuming ozone, then they can't be leaving the canopy. What is needed here are some simple box model calculations with assumed residence/mixing times below and within the canopy and with some simple SQT/O₃/OH chemistry to explore whether or not the modelled SQT flux rates from the soil can in fact (1) account for low observed ozone and (2) escape from the canopy without prior chemical degradation.

lines 167-9: without these box model calculations it is not possible to make the speculative comments in lines 167, 168 and 169.

169-171: and without these box model calculations it is not possible to make the even more speculative comments made in 169, 170 and 171. Deposition rates of particles within the canopy are extremely high and so some validation of the speculation that new organic particles formed by SQT/O₃ reactions close to the forest floor (below and within the canopy) can themselves escape the canopy to become CCN in the atmosphere is needed.

Overall summary:

This is a very interesting manuscript. The measurements and model results combined make a compelling case for forest soils acting as an important and significant source of SQTs, and this is definitely worthy of publication. The speculative comments made in the discussion (line 162 onwards) are very interesting and potentially important, but they contain a possible inconsistency that simply must be resolved. SQTs are highly reactive and so may remove ozone from below the canopy - but in that case those SQT molecules cannot escape the canopy into the atmosphere. It may be that the emission rate from the soils is sufficiently large that some SQT molecules may react with ozone in sufficient quantity to explain the observed low ozone concentration ratios, while leaving enough SQT molecules to escape the canopy at rates that rival emissions from the trees themselves. But before these speculations can be published they need exploration to show whether or not they are at least possible or feasible - at the very least using a simple forest canopy box model (such as the one developed by my colleague Kirsti Ashworth, but I am sure there are others as well). An alternative approach would be to remove these speculative comments - but that would be a shame, as they are very interesting and potentially important.

Nick Hewitt, Lancaster University

Reviewer #3 (Remarks to the Author):

Comments to the manuscript "Strong sesquiterpene emission from Amazonian soils" by Bourtsoukidis and colleagues.

The present manuscript deals with an interesting and actual topic of research. It becomes more and more clear that in soils bacteria and fungi are a biogenic source of VOCs, in particular the semi-volatile sesquiterpenes (SQTs), that have been overlooked in the past when research concentrated on leaf level (stem) and canopy VOC fluxes. Due to the high reactivity of SQTs with atmospheric oxidants (OH radicals and ozone) it is very likely that these compounds getting lost during analysis, i.e. under field conditions when air composition cannot be controlled accurately as in the laboratory.

The present manuscript describes laboratory experiments with tropical soils from Amazonia and surrounding areas demonstrating SQT emission. By controlled drying out cycles under different atmospheric conditions (anaerob and aerob) the SQT emissions were parameterized and used to develop an emission algorithm enabling the calculation of SQT fluxes from the selected tropical soil types. Based on this algorithm a model exercise was performed comparing the SQT fluxes from canopy and soils at the ATTO site in Brazil.

The manuscript is very interesting describing novel findings. Overall the work shows that SQT emissions from tropical soil are highly variable depending on the soil type, with highest emission rates found in soil samples from "Tera firme" forest. It's also interesting to note that these emissions originate from bacteria but not from fungi (at least in the measured soils) indicating different microbial activities. This is also obvious for the large variation of SQT emission from the TF samples 1-3. Despite their high degree of similarity between physiochemical properties the emission rates differed strongly.

This brings me to the critical points of the manuscript. The data are novel in the sense that they describe soil SQT emission from a globally important ecosystem. However, I have doubt that based on the current data a solid soil emission model can be drawn. The model was fit with the current data and also validation was done with the same samples. Therefore, it is not astonishing that the experimental data and modeling output fit. As seen in Fig 3c, the overlay of field data and the modeled data is not as clear. The comparison of modeled canopy and soil SQT fluxes shows a potentially very high contribution of SQTs from the soil to the overall SQT flux. Keeping in mind the soil surface area relative to the much larger leaf surface per ground area in the canopy (MODIS LAI for Amazonian rain forest from c. 4-6 m² m⁻²; Doughty and Goulden, 2008 JGR) the SQT emissions from topical soils are very high. This has to be brought in context with the few published reports on soil SQT emissions. Overall, the discussion is very weak (condensed), focusing on air chemistry rather on the possible biology behind the SQT emissions from soil.

Beside this shortcoming I see some additional points, which should be improved in the different sections of the manuscript i.e. in the presentation of the data.

Methods

For the reader it is very difficult to follow the numbers S1 to S? On line 402 it is stated that 38 samples were measured. In maximum I have seen data up to S25 (Extended data figure 2. So where is the rest? You should follow a clear layout e.g. according to the soil samples and soil layers.

Line 401ff: The cuvette system uses an active flow through the soil sample. That's not the case in nature. Under natural conditions movement of gaseous compounds are driven in soil by diffusion and not an active flux. How do you think your experimental set up has influenced the SQT emissions rates? It might be that the emissions are high because of this active gas stream. Please discuss this point with respect to your field system and published data where the incoming cuvette air streams over the soil and not through the soil pores.

Line 462: E/N is changing strongly when drift tube pressure changes from 2 to 2.2. mbar. Please give the range. As mentioned in line 477 the fragmentation pattern strongly depends on E/N. Hence it is important how strongly your E/N differs.

line 564 and related Fig3. How have you measured soil emission fluxes when a rain event occurred? Have you removed you soil chambers during the rain fall and put them back later?

Figures are highly complex and in some cases hard to decipher:

Figure 1:

Overall, this figure is rather complex. It would benefit from a re-organization e.g. SQT pattern and SQT legends separately from the curves. Don't see the need of the background image.

In (a) what does the blue line represents? I assume you show monoterpenes. What does the shaded area in (a) represents? In the legend in Fig 1e the asterisk indicates that the subsoil was measured under aerobic conditions. Why? Why now showing the regular anaerobic situation? How many n are measured?

Figure 2:

Legend: which soil samples were shown O, A or B? Please indicate.

Figure 3 and general comment:

It is not astonishing that modeled curves fit to experimental data when the later are used as input parameters for the 2 submodels (HM and MM). How does the general model perform for the other soil samples? Have you tested / validated your algorithm with other samples not used for parameterization?

Fig 3a: the grey symbols mean aerobic and not anaerobic.

Fig 3b: no SQT measurements under anaerobic are given for the topsoil (a). Does it mean that in this layer at aWFPS of e.g. 90% the conditions are aerobic while in the O horizon at the same WFPS the conditions are anaerobic?

Fig 3c: what does the left part with the box ? Was this TF1 sample measured a day 2? Remove for clarity.

Figure 4:

Legend: "Daily averages of modeled SQT emissions...." How does the data look like for TF2? What's with this sample? It's barely mentioned in the paper (except extended data figures 1 and 2). Why haven't you used all samples from the ATTO side to show the high variability of soil SQT emissions.

Extended data figure 1:

How you define outliers? It seems they only occurred in the 0-5 cm horizon O.

At which WFPS you consider the samples to be anaerobic? Please define.

Extended data figure 2:

Please indicate /mark which sample number belongs to which soil sample. Here you mention for some samples WFPSs of c. 45 % defining them as anaerobic.

In Figure 3a you classified anaerobic conditions down to a WFPS of c. 90% in the O layer. In Fig 3b no anaerobic conditions at all, though the WFPS was above 90%. So what's true? At which WFPS you have a change from aerobic to anaerobic conditions? Please be consistent throughout the manuscript.

Extended data figure 3:

Which soil layers are used for this comparison? In(c) you show much more data points as in (a) and (b). Where do they come from? SQT emissions from PTR-MS data were calculated from m205. Please indicate.

Reviewer #1 (Remarks to the Author):

General comments

Reviewer#1 comment: This study examined the contribution of neotropical soils to the atmospheric charge of biogenic volatile organic compounds in the Amazonian basin. In well-designed laboratory experiments the influences of several environmental factors such as soil moisture, temperature and atmospheric composition on sesquiterpene emissions were investigated on a large number of soil samples subdivided in diverse horizons. Based on the results the authors developed an emission model, which was further integrated in an existing modelling framework for spatial and temporal extrapolation. In addition field measurements were conducted as well as specific RNA analyses of soil samples were to elucidate the biological origin of soil VOC exchanges. In addition RNA analyses of soil samples were conducted to elucidate the biological origin of soil VOC exchanges. Overall the results provide strong evidence that soil bacteria emit large amounts of sesquiterpenes representing a large, hitherto overlooked source of highly reactive carbon.

The results are novel and are important to scientists working in the field of atmosphere biosphere interactions. Research on VOC exchange at the soil atmosphere interface has received increasing attention during the last decade, but comprehensive studies such as the present one are sparse.

The paper is overall well written and should make a nice contribution to Nature Communication. However, prior publication some clarifications and corrections are needed.

Reviewer#1 comment: I have been a bit disappointed by the section Discussion, which is very brief. In fact it resembles rather a listing of conclusions than a critical examination of the study in the context of the current understanding of soil VOC exchanges. On the other hand, in other sections the description of the results and methods contain many statements of data interpretation that could be moved to the section Discussion (e.g. L66 ff; L583-586). Alternatively the current section "Discussion" is maintained as it is but renamed as "Conclusions". In any case there are several directions in which the authors could have more extended the discussion. I hope my following comments listed below will help doing so.

Response by authors: We agree with the reviewer that the discussion section can be expanded and as the Nature Communications editor has noted, space is available. We therefore now follow the suggestion by the reviewer and include an expanded discussion section, renaming our former discussion as "conclusions".

Reviewer#1 comment: I wonder whether the authors performed statistical analyses to check whether emissions are significantly different between soil types and horizons, sites, environmental conditions a.s.o.(see e.g. description of results L84-92)

Response by authors: All data and available statistics between soil layers and aerobic vs anaerobic conditions are now presented in the main text (new Fig. 2 and 3). In Fig. 2 we display the mean and the 25th and 75th percentile for each soil horizon. In addition, we present all data for each site with the error bars at each data point to illustrate the standard deviation of the replicates (summarized in Ext. data Table 1). In Fig. 3 we clearly show all measured emissions for top soil for each site and environmental condition. Given the fact that there are counter-influence parameters (soil type, depth, oxygen, temperature), the p values derived from significance tests would be depend on relatively few samples and therefore it would not be a robust measure. To avoid confusion, we have additionally replaced the word "significant" with "important".

Reviewer#1 comment: The authors mention several times in the ms (e.g. L426 ff) that for simultaneous quantification of gas uptake and production rates low and atmospherically relevant VOC mixing ratios were introduced to the chamber air supply during laboratory experiments. However no precisions are given about the exact VOC amounts and composition that were applied during laboratory fumigation experiments and no results are given about uptake, remission or transformations of fumigated VOC.

Response by authors: We are somewhat confused with this comment since both the VOC amount (L428: 0.1-1ppb) and uptake results (Fig. 1; monoterpenes) are presented. In addition, we briefly discuss the uptakes as following:

“Contrasting behavior was reproducibly exhibited by methanol, which was weakly uptaken under wet conditions and released during drying, while monoterpenes were (in contrast with the canopy) weakly emitted under wet conditions and moderately consumed in the low moisture range.” For clarity, we now include *“14 VOC components; Apel-Riemer Environmenta, USA diluted to the 0.1-1 ppb range”* in the methods section and in the sentence commented by the reviewer (former L428).

Reviewer#1 comment: RNA analyses strongly suggest that soil bacteria are the main sources of SQTs in the present study. Unfortunately these analyses were made only on TF4 and TF5 soils of the dryer outer region of the Amazonian basin (Cerrado-savannah ecosystem) and not on soils of the inner tropical rain forest, on which however the authors concentrated their emission studies (cf table 1). This limitation is not explained nor discussed in the manuscript.

Response by authors: This is a fair point and we are aware of this limitation. Our aim was to identify specific sources for the SQT emissions from soil, and therefore we performed a joint measurement of SQT emissions and rRNA analysis for samples from the Amazonian basin where we expected a high contrast in both SQT emissions and rRNA analysis based on previous annual fire treatments and control. This approach is now explained as follows along with an acknowledgement of the limitation in generalizing these results to TF4 and TF5:

“Despite the limitations that may arise from the use of rRNA as an indicator of microbial activity, both the ecological history and rRNA dynamics observed indicate that the microbial activity drive the SQT production and release for these soils. TF4 and TF5 belong to a dryer region of the Amazonian basin, and therefore the bacterial/fungal contribution to the net SQT production in the entire Amazon basin requires further investigation.”

Reviewer#1 comment: In the lab soils were assayed under anaerobic conditions were simulated by incubation in pure Nitrogen. I guess that under natural conditions proxy anaerobic conditions (hypoxia) in the upper soil horizons will exclusively caused by water logging (water saturated soils) after flooding and heavy rainfalls. If so results from laboratory experiments under dry anoxia conditions might not really represent common conditions in nature?

Response by authors: The reviewer is correct. As our hydrological model suggests such conditions only briefly (1-2 hours) occur after heavy rainfall. This is the reason that our SQT emission model does not account for anaerobic conditions apart from the cases when the WFPS is >80%.

Reviewer#1 comment: The laboratory work concentrated on soil exempted from roots. While I understand the author’s choice, it represents a potential limitation since the rhizosphere with associated microbes may strongly affect VOC exchange characteristics of soils. I guess that in dense forests with fine roots are everywhere.

Response by authors: This potential limitation is now discussed for clarity. In direct response to this comment, we did not include roots in our experiments because a) we would not have been able to

distinguish emissions from soil and roots, b) cut or damaged roots would emit very different amounts and species of VOC. The respective text part now reads as:

“The presence of severed or damaged roots in the samples analyzed, may induce unnatural emissions from the rhizosphere that would lead to difficulties in distinguishing soil and root emissions. Therefore, all fine roots were carefully removed from the samples, prior to their analysis. Potential remaining fine roots may increase the PTR-MS signal but their potential abundance (<0.1%) is not expected to induce the release of VOCs substantially.”

More specific comments and questions:

Reviewer#1 comment: L40: I suggest replacing “aid” by “affect”.

Response by authors: Replaced as suggested.

Reviewer#1 comment: L77-78: “ Acetone emissions were the strongest, with release rates up to 1.5 mg m² h⁻¹; an order of magnitude higher than acetone emissions from Amazonian vegetation²⁶.” For the sake of clearness I suggest replacing “...than acetone emissions from Amazonian vegetation²⁶” by “...than acetone net fluxes measured above an Amazonian forest²⁶”.

Response by authors: We thank the reviewer for the suggestion. The text was modified accordingly.

Reviewer#1 comment: Similarly L82: please insert “fluxes” after “canopy”.

Response by authors: Inserted as suggested.

Reviewer#1 comment: LL84-94: The differences in SQT emissions that are described in this paragraph are not backed by results from statistical analyses.

Response by authors: All data and available statistics between soil layers and aerobic vs anaerobic conditions are now presented in the main text (new Fig. 2 and 3). In Fig. 2 we display the mean and the 25th and 75th percentile for each soil horizon. In addition, we present all data for each site with the error bars at each data point to illustrate the standard deviation of the replicates (summarized in Ext. data Table 1). In Fig. 3 we clearly show all measured emissions for top soil for each site and environmental condition. Given the fact that there are counter-influence parameters (soil type, depth, oxygen, temperature), the p values derived from significance tests would be depend on relatively few samples and therefore it would not be a robust measure. To avoid confusion, we have additionally replaced the word “significant” with “important”.

Reviewer#1 comment: L413: I do not understand why “Potential remaining fine roots may interfere with the PTR-MS signal”?

Response by authors: Similar to our response above, the objectives of our study are to investigate soil emissions and therefore the presence of damaged or severed fine roots may increase/decrease the signal of our instrumentation. VOC emission work on plant branches and leaves above ground has established that damaged plants emit very differently (in species and strength) to when measured in the natural state. The same effect was anticipated for root systems which are inevitably damaged during the extraction of the soil sample. Here our focus is on soil emissions and so to simplify the analysis we adopted the procedure described to remove the roots from the samples. This procedure is commonly used in soil science analysis. We also refer you to our previous response and revisions made.

Reviewer#1 comment: L113-114: “Along with the production strength, the chemical diversity of SQT emissions increased. Possibly this is just a matter of detection limit?”

Response by authors: We are confident that this is not the case. The chemical diversity was investigated by GC-MS, with very low detection limits and therefore any change in individual sesquiterpene composition would have been visible in the chromatograms. The production strength can be seen by comparing Fig1b and Fig1c and the respective chemical diversity in the respective pie charts. It is clearly visible that several species quantified under anaerobic conditions were not present in the chromatograms analyzed for the aerobic conditions. We note that all samples were treated equally and analyzed in the same time frame and methods and that *“a total of 10 different SQT were released and the emission ratios have changed markedly”* which indicates different processes involved.

Reviewer#1 comment: L123. “represent common conditions in nature” see my comment above.

Response by authors: The reviewer probably refers to the anaerobic conditions that are not common for these soils. As explained above, the soils may become anaerobic for short periods after heavy rainfall as the water percolates down through the surface soil. Therefore our model was built to incorporate this effect and the VOC emissions under these conditions were measured. Please note that anaerobic conditions may occur for just few hours after the rainfall and the normal conditions are almost always in the moderate moisture (MM) regime. Although the time of anaerobic conditions is short, the emission under these conditions is strong. Therefore, we would like to keep the sentence as is in the manuscript, since the laboratory experiments do “represent common conditions in nature” and they were designed with this intention. An excellent example can be seen in the field experiments (Fig. 3c) where a strong rainfall (and absence of rain for few days) simulated exactly the laboratory conditions. In addition to the above, please note that the model run with aerobic condition only (Fig 3c, dashed red line) underestimates the field measurements. Hence, we came to the hypothesis anaerobic mechanisms were activated. This is now discussed in the respective section.

Revised for clarity as: *The emissions observed in the laboratory are the result of conditions commonly occurring in nature. A natural rain event initiates a cascade of physiochemical and microbial processes as the water percolates through the soil layers.*

Reviewer#1 comment: L133 What are “optimum emission functions”?

Response by authors: “Optimum emission functions” are all the experiments where the optimum shaped emission pattern was observed (e.g. Fig 1). This sentence is intended to demonstrate the reproducibility of the experiments. For reasons of clarity we now elaborate further on this sentence: *“The emission model algorithm was applied in all laboratory experiments where an emission optimum was observed for a particular WFPS. A close agreement between the emission algorithm and measured emissions ($0.89 < R^2 < 0.97$) indicates that the algorithm can reliably simulate the observed SQT emissions (see example in Fig. 3a and 3b)”*

Reviewer#1 comment: L168: I suggest replacing “will” by “may” because the interference of ozone in VOC mediated plant-insect interactions have yet been demonstrated only by a limited number of studies.

Response by authors: Changed accordingly.

Reviewer#1 comment: Figure 1 and caption.

Fig 1a: L198-199: “a. Normalized soil fluxes of SQTs (red) and acetone (purple) as a function of WFPS. The normalized, algorithm derived emission curve is the result...”
normalized fluxes with respect to what? What is the blue line in 1a?

Response by authors: Normalized to the optimum emission observed at optimum WFPS. The blue line in Fig1a represents acetic acid. We have now removed acetic acid for clarity and the figure's caption has been modified accordingly.

"The normalized (to the optimum emission observed at moderate moisture) algorithm derived emission curve is the result of integrating laboratory observations from...."

Reviewer#1 comment: LL 205-206: "The asterisk in panel e denotes that the experiments were performed under aerobic conditions." If measurements were made under aerobic conditions why in Fig 1e anaerobic is mentioned?

Response by authors: Because at such depths it is common to meet anaerobic conditions. However due to experimental limitations (i.e. restricted N₂ supply), the experiments were conducted under aerobic conditions. The purpose of Fig. 1e is to demonstrate the absence of VOC emissions in deeper soils.

Addition in the figure's caption:

"The asterisk in panel e denotes that the experiments were performed under aerobic conditions despite the predominant anaerobic conditions in the deep soil layers. Note that VOC emissions at this depth are very low"

Reviewer#1 comment: L277: What is meant by "...and is between area comparison parameter"

Response by authors: On reflection we consider this sentence unnecessary. It was just intended to emphasize that we are comparing the soil types only from the first 10 cm and not by the complete profile. We have removed this phrase for clarity.

Reviewer#1 comment: Extended data table 2: please add standard deviations or standard errors to the data where applicable (for N>1). Further I suggest showing absolute emission rates as well (+/- Stdev), for example in parenthesis below each percentage data.

Why RTs are not given for all compounds?

Response by authors: We thank the reviewer for the comment. Stdev is now added. Absolute emission rates are continuously quantified at high frequency by the PTR-MS and a comparison between PTR and GC measurement techniques is given in Ext.data Fig. 3. Therefore absolute values of just the GC are not added to the table.

It's not the retention times (RTs) that are presented in Table 2 but the retention indexes (RIs). The first compounds were identified with authentic standards and the rest by using the RIs (as described in the methods section).

In the process of including the Stdev we have noticed an averaging error which is also now corrected.

Reviewer#1 comment: Extended data figures 1 and 2:

Increase size of symbols

According which criteria the authors defined data as outliers?

Response by authors: The symbols have been increased and according to the software's definition (matlab), "The boxplot draws points as outliers if they are greater than $q3 + w \times (q3 - q1)$ or less than $q1 - w \times (q3 - q1)$. $q1$ and $q3$ are the 25th and 75th percentiles of the sample data, respectively. The 'Whisker' corresponds to a $\pm 2.7\sigma$ and 99.3 percent coverage if the data are normally distributed. (<https://de.mathworks.com/help/stats/box-plots.html>)".

By reconsidering the comment of the outliers in this specific case (Ext. data Fig. 1), we have now removed the outliers. Nonetheless, we kept the outlier data in Ext. data Fig 2 as they help show all points during each experiment.

Reviewer#1 comment: Extended data figure 3: Is this figure cited in the text?

Data in Figure C are shown in a very “compressed” manner. I suggest increasing the height of this figure and showing data in a normal scale in addition to the logarithmic scale. The reliability of the identification and quantification of SQTs by PTR-MS (see also L462 ff) is crucial in the present study, because the postulated high emission quantities present the most original outcome of the work. In my view a consistent quantification of PTR-MS for SQTs requires regular calibration with gaseous SQT-standards based on and controlled by GC-MS measurements made in parallel at least once per experiment on the different soil samples. Apparently no SQT standards were used for PTR-MS calibration and results from parallel GC-MS measurements are not presented for all soil samples (cf extended data table 2).

Response by authors: This figure was not cited in the text and we thank the reviewer for noticing it. The comparison presented included all the lab data in table 2. It does not include the field data depicted in Fig.3 (there were no PTR-MS measurements in the field). Possibly the confusion occurred from the fact we did not include TF2 in the table (N=2). Now both Table 2 and Ext.dataFig3c show the N=27 absorbent tube samples.

We agree with the reviewer that quantification is crucial. As no gaseous SQT standards are available as these compounds are too involatile and sticky to produce a stable pressurized gas mixture, we have adopted a theoretical approach to the PTR-MS data quantification that has proven reliable in previous works (described in detail in the methods section). Interestingly, comparing the GCMS and the PTRMS measurements show that the PTR-MS slightly underestimates the reported values in the highest emission range.

Our aim with Ext.dataFig3c is to show the close agreement between PTR-MS measurements and GC-MS measurements. It becomes clear that the large magnitude span of the measurements, makes it difficult to properly depict the generally good correlation between the two measurement methods for the smaller emission rates. Therefore, we have now modified the figure to “showing data in a normal scale in addition to the logarithmic scale” and changed the caption to:

“In d, the same points are plotted on a linear scale to in order to make the slight underestimation of the PTR-MS measurements visible”.

Reviewer#1 comment: L423-424: I do not understand how field bulk density data can be applied to sieved soil samples?

Response by authors: We recognized this limitation and hence we tried to be as close as possible to the field conditions when putting the sieved soil inside the chambers. However, once the soil is dried and sieved the lab bulk density cannot be determined in a standardized way. Therefore, we have chosen to adopt the commonly used method (e.g. Meixner et al.,2006) to use the field bulk density as characteristic of the site. We added the following text:

“While the laboratory soil density can be slightly different after sieving, the field bulk density was chosen as characteristic for each soil type to be consistent with previous studies. The directly measurable quantity of field bulk density was therefore used to derive gravimetric soil moisture in this work.”

Reviewer#1 comment: L434: what was monitored?

Response by authors: We are somewhat confused with this comment. Maybe the reviewer wants us to specify that VOCs were measured inside each chamber? We had 6 soil chambers operating in each experiment and the valves were switching to the next chamber every 10-15 minutes. For clarity we revise this sentence as following:

“The air from the outflow of each chamber was monitored by the PTR-MS for multiple VOCs including sesquiterpenes. Each cycle was 10-15 min before switching to the next chamber. ”

Reviewer#1 comment: L467: please explain “experimentally derived transmission curve”.

Response by authors: Explained as following:

“(i.e. while calibrations were used for the species in the calibration standard, SQTs were calculated by applying the instrument’s transmission curve which is a commonly used procedure for species without stable calibration sources. The transmission curve takes into account the changes in ion transmission efficiency through the detector which change with size.)”

Reviewer#1 comment: L481: The brand name and model of the second PTR-MS system should be quoted.

Response by authors: They were different systems but the same model. This information is now included in the text.

“...were conducted with a different but same model quadrupole PTR-MS system (Ionicon analytik GmbH, Austria)...”

Reviewer#1 comment: L486: This statement contradicts what is stated in line 481 (S3, S4, S13, S15, S16 were conducted with PTR-MS2).

Response by authors: We thank the reviewer for noticing this typographic mistake. The confusion occurred due to the falsely stated “(using PTRMS1)” which is now removed.

Reviewer#1 comment: L587-590. The statements here are somewhat confusing. Adsorbent tubes of field measurements were analysed in Finland but not those of the laboratory experiments? Where were the laboratory samples analyzed? All were analyzed with exactly the same analytical set-up and calibrated with the same authentic standards?

Response by authors: All adsorbent tubes were analyzed by the same instrument and with the same method. In the lab experiments (performed in Brazil as soil export was restricted): “Upon collection, the adsorbent tubes were shipped to Finland and analyzed between 3-4 weeks later”.

Once we recognized the importance of having field confirmation on our model, a-gurjunene was purchased by the Finnish group and the field sampled adsorbent tubes were analyzed using an authentic liquid standard for a-gurjunene in order to confirm the RT defined by the RI mentioned in

table 2. It was shown that the RT identified by the RI was correct.

Reviewer#1 comment: Additional references for consideration:

Leff JW, Fierer N. 2008. Volatile organic compound (VOC) emissions from soil and litter samples. *Soil Biology and Biochemistry* 40: 1629-1636.

Schade GW, Custer, Thomas G. 2004. OVOC emissions from agricultural soil in northern Germany during the 2003 European heat wave. *Atmospheric Environment* 38: 6105-6114.

Asensio D, Peñuelas J, Filella I, Llusà J. 2007. On-line screening of soil VOCs exchange responses to moisture, temperature and root presence. *Plant and Soil* 291: 249-261.

Final remark by authors: We are grateful for the detailed and comprehensive review by the reviewer#1. We believe that we have addressed all the comments and suggestions made.

Reviewer #2 (Remarks to the Author):

This manuscript addresses an important topic in current biogeochemistry - how do the forests (and soils) of Amazonia interact with the atmosphere and what are the Earth system implications of these interactions? More specifically, very considerable research effort has been expended on understanding how the Amazonian forest contributes reactive hydrocarbons and other organic compounds to the atmosphere, and what the impacts of these compounds are on regional-scale atmospheric chemistry. Reactivity measurements have indicated that there is likely an additional strong source of VOCs in the forest environment, other than trees themselves. In this manuscript it is shown that forest soils may be a very considerable source of reactive sesquiterpenes to the atmosphere, at emission rates which match those from the forest canopy. This is an important and novel finding which will be of interest to a wide readership which is worthy of publication in *Nature Communications*. However, I do have one very major comment/reservation that does require attention before the manuscript can be accepted for publication - that relating to the discussion - from line 162 onwards - see below.

Specific comments that require responses:

Reviewer#2 comment: lines 32/33: quoting modelled flux rates to one tenth of a microgram/m²/h gives a spurious sense of precision which is completely unjustified. On what basis does a value of ~100 "rival" one of ~170? On what basis is the quoted degree of uncertainty estimated?

Response by authors:

The main message of the paper is that sesquiterpene emissions from soils are significant and under certain conditions of comparable magnitude forest canopy emissions. The use of the word "rival" was meant to draw attention to the fact that these emissions are of comparable magnitude and have been previously overlooked. From the reviewer's comment we realize that we should make this statement more precise by mentioning the uncertainties in the estimate, and soften the statement to state "comparable magnitude".

Regarding the degree of uncertainty, the values reported here are averages over the dry seasons for both tree emissions and our strong soil emitter (TF1). The mean values and the std devs reported are the product of daily values as shown in Fig. 3.

One also has to consider the even larger uncertainties in the latest forest emissions. The emissions model MEGAN includes a temperature dependency (beta factor) of 0.17 for SQT emissions for

Amazonian vegetation. However, to the best of our knowledge there is no study that determined this beta-factor from Amazonian vegetation and therefore the current emission models are still based on a generalized formula. As can be seen in Bourtsoukidis et al. (2012) this beta-factor can vary significantly day-to-day and with environmental conditions, at least for *Norway spruce*.

The aforementioned points are now added and the text reads as follows:

“The comparison with an established emission algorithm for vegetation (MEGAN) indicated that soil emissions from TF1 can rival the canopy emissions at the same location, being of comparable magnitude in the dry season. The inherent uncertainties included in this comparison primarily originate from the temperature dependency (β -factor) that is used for vegetation emissions, and the large variability that has been observed for the soil emissions. To the best of our knowledge, no studies have directly addressed and determined the temperature dependency of SQT emissions from Amazonian vegetation. Hence, the present implicit assumption of a common and stable season independent β -factor =0.17 (as used in MEGAN for all SQT species) can be considered a rough approximation with large associated uncertainty. It has been shown that the β -factor can vary significantly day-to-day, and with atmospheric conditions (Bourtsoukidis et al., 2012). In addition to canopy model uncertainties, soil emissions displayed significant variation in strength between the sites investigated, with emission estimates ranging from a few $\mu\text{g m}^{-2} \text{h}^{-1}$ to orders of magnitude higher. In general soil emissions seem to be primarily location and microbial activity specific, as demonstrated by TF4 and TF5.”

Reviewer#2 comment: Fig 1: What is the replication here? Are these plots from single measurements? What does the shading represent?

Response by authors: The results depicted in Fig.1 are the product of 4 separate 1 week dry out experiments (b-e) and their modelling output (a). Our intention was to provide a comprehensive overview profile of SQT emissions from the different soil layers and illustrate the optimum shaped function of SQT emissions from soils. In this figure, the shading represents the standard deviation from each measurement point in the lab (see L434). In order to be clearer, we have included the following information in the caption of Fig. 1:

“ The shaded areas indicate the standard deviation of the measured emissions rates at each chamber cycle”.

Reviewer#2 comment: Fig 2 contains more information than this reviewer can process in one diagram. I am not suggesting that the figure has to be changed, but the authors should at least consider alternative ways of displaying this information. What is the degree of replication in this figure?

Response by authors: We have had extensive discussions on the best way to display these data and results. We agree that the current approach does include a large amount of information but on the other hand, by condensing all information in the same figure can give the reader a good overview. One approach would have been to merge the TF4 and TF5 emissions but these data have been acquired with different experimental setups. Another approach would be to omit one of the samples (eg. S13 and S16) but we again believe that the replication information is vital here (also given the different time resolution on the same-soil experiments).

Reviewer#2 comment: Fig 3 caption puts forward an explanation for the observed behavior in terms of two processes - but this is purely speculative, and should be expressed as such. No indication of the degree of replication is given.

Response by authors: The assumption of the two process model is now clearly stated in the discussion. The degree of replication was mentioned in the respective text (L132-133). Nonetheless, we understand the confusion that may arise from such condensed sentences and we have revised accordingly.

HM and MM regimes: *Strong sesquiterpene (SQT) emissions from Amazonian soils have been quantified from laboratory investigations. A reproducible emission pattern as a function of water filled pore space (WFPS) was recognized and an empirical emission algorithm was created assuming different microbial processes under two different moisture regimes (high moisture/initial burst HM and medium moisture/drying out MM).....Followed by a detailed discussion for the two regimes (see discussion section).*

Degree of replication: *“The emission model algorithm was applied in all laboratory experiments where an optimum emission at a certain WFPS was observed (i.e. organic (O) horizon topsoil (A)). A close agreement between the emission algorithm and measured emissions ($0.89 < R^2 < 0.97$) indicates that the algorithm can reliably simulate the observed SQT emissions (see example in Fig. 3a and 3b)”*

Reviewer#2 comment: line 64 onwards: no indication of experimental replication is given. Where these single experiments or was there true experimental replication?

Response by authors: As we indicate in this sentence, the burst released “was observed for the majority of the incubated soils”. There was true experimental replication of all single experiments (e.g. TF1 soil was repeated 4 times). To make it clearer for the reader we denote that the majority means (>80%) of the samples investigated. Unfortunately we cannot take into account all 32 samples targeted by this expression (S1-S22, S26-S35) since a few experiments were started long after the initial water pulse due to experimental problems. This 80% represents the 27 samples that were performed with the addition of the water in-time with the beginning of the experiment. In 22/27 of these experiments the burst was observed.

Reviewer#2 comment: line 96-97: is the transcript abundance expected or assumed to vary linearly with bacterial/fungal activity? A little more explanation of why the transcript were measured and what assumptions are being made about them is needed here.

Response by authors: Transcript abundance does not always vary linearly with activity, as it has been shown for some taxa in which the amount of ribosomal RNA is not directly correlated with activity during growth curve (Molin and Givskov, 1999). Dormant cells can also contain rRNA, meaning that transcript abundance can vary other than only to microbial activity (Blazewicz et al., 2013). On the other hand, ribosomes indicating potential cell activity in a community are more abundant than in dormant cells (Kerkhof and Kemp, 1999) and hence the use of molecular tools has been practical in the last decades to enable the study of microbial community responses to the environment variables (Barnard et al., 2013; Klein et al., 2016).

The following text has been added in the discussion section:

“Under the MM regime, SQTs displayed optimum emissions at certain WFPS reproducibly, thus indicating the most favorable conditions of microbially produced SQTs. The monitoring of 16S- and 18S- rRNA transcripts was undertaken as a practical means to study microbial community responses to environmental variables (Barnard et al., 2013; Klein et al., 2016). While some taxa do not demonstrate a linear correlation between rRNA abundance and microbial activity (Molin and Givskov, 1999), ribosomes indicating potential cell activity in a community are more abundant than in dormant cells (Kerkhof and Kemp, 1999). At the community level, an increase in rRNA content is assumed to reflect greater protein production over time, and therefore a proxy for activity. Our experiments

demonstrated a variation of transcript abundance between two ecologically diverse soils, and in addition, they showed a peak of transcription coincident with SQT peak emissions in the MM regime. This strongly suggests that microbial activity inferred from ribosomal can be associated with SQT production and release.”

Reviewer#2 comment: lines 162/3 and lines 166/7: There is a nasty inconsistency in the arguments made here. On the one hand, the authors are claiming that SQT emissions from the forest soils may leave the forest canopy into the atmosphere - at rates that rival those from the trees themselves. On the other hand, they then say that SQTs emitted from the soils may react below and/or within the canopy, so explaining low observed ozone levels. Well, they can't have it both ways so easily....if SQT molecules are consuming ozone, then they can't be leaving the canopy. What is needed here are some simple box model calculations with assumed residence/mixing times below and within the canopy and with some simple SQT/O₃/OH chemistry to explore whether or not the modelled SQT flux rates from the soil can in fact (1) account for low observed ozone and (2) escape from the canopy without prior chemical degradation.

Reviewer#2 comment: lines 167-9: without these box model calculations it is not possible to make the speculative comments in lines 167, 168 and 169.

Reviewer#2 comment: 169-171: and without these box model calculations it is not possible to make the even more speculative comments made in 169, 170 and 171. Deposition rates of particles within the canopy are extremely high and so some validation of the speculation that new organic particles formed by SQT/O₃ reactions close to the forest floor (below and within the canopy) can themselves escape the canopy to become CCN in the atmosphere is needed.

Response by authors: These are excellent (and in the same line) comments that could have been addressed only with the incorporation of a forest canopy box model. Following the reviewer's suggestion we have now included such model and as the results indicate, soil SQTs dominate the O₃ reactivity at the forest floor and do play an important role on ozonolysis even at the canopy top, during nighttime. We have now included these results and the text that reads as following:

The average dry season SQT emissions from Terra Firme soils (44.9 $\mu\text{g m}^{-2} \text{h}^{-1}$ considering emissions from TF1, TF2, TF3 and TF4) were incorporated into a simple forest canopy column model in order to evaluate the significance of soil SQTs to air chemistry. The majority of these highly reactive species were lost through ozonolysis within the canopy space, with a maximum of 1.5% of soil emissions escaping the canopy just before dawn when photochemistry commences and around 0.2% through most of the day as reaction rates reach a maximum. Soil emissions of SQTs account for as much as 50% of O₃ reactivity (soon after dawn) at the soil surface and a relatively constant 30% during daylight hours. Overnight, soil SQTs contribute nearly 40% of O₃ reactivity at the top of the canopy, but this is substantially reduced to 0.5-1% during daylight hours as the SQTs are rapidly consumed within the canopy. The high reactivity of SQTs results in substantial enhancements in the concentrations of condensable reaction products that will partition into the aerosol phase and potentially act as CCN. SQT-derived oxidation products increase the total concentration of condensable species at the top of the canopy by 30-40% at night when reaction rates are low for other emitted compounds decreasing to around 20% at dusk when the contribution from other species is highest. These pilot model runs suggest that not only do emissions of SQTs from Terra Firme soils substantially affect atmospheric chemistry within the canopy but that the effects have the potential to alter regional chemistry-climate.

Reviewer#2 comment: Overall summary:

This is a very interesting manuscript. The measurements and model results combined make a compelling case for forest soils acting as an important and significant source of SQTs, and this is

definitely worthy of publication. The speculative comments made in the discussion (line 162 onwards) are very interesting and potentially important, but they contain a possible inconsistency that simply must be resolved. SQTs are highly reactive and so may remove ozone from below the canopy - but in that case those SQT molecules cannot escape the canopy into the atmosphere. It may be that the emission rate from the soils is sufficiently large that some SQT molecules may react with ozone in sufficient quantity to explain the observed low ozone concentration ratios, while leaving enough SQT molecules to escape the canopy at rates that rival emissions from the trees themselves. But before these speculations can be published they need exploration to show whether or not they are at least possible or feasible - at the very least using a simple forest canopy box model (such as the one developed by my colleague Kirsti Ashworth, but I am sure there are others as well). An alternative approach would be to remove these speculative comments - but that would be a shame, as they are very interesting and potentially important.

Nick Hewitt, Lancaster University

Final remark by authors: We thank Prof. Nick Hewitt for his comprehensive and critical review. The addition of the forest canopy box model does complete our study in an important and conclusive way and hence we are very grateful that this review led to the implementation of such box model in our study.

Reviewer #3 (Remarks to the Author):

Comments to the manuscript "Strong sesquiterpene emission from Amazonian soils" by Boursoukidis and colleagues.

The present manuscript deals with an interesting and actual topic of research. It becomes more and more clear that in soils bacteria and fungi are a biogenic source of VOCs, in particular the semi-volatile sesquiterpenes (SQTs), that have been overlooked in the past when research concentrated on leaf level (stem) and canopy VOC fluxes. Due to the high reactivity of SQTs with atmospheric oxidants (OH radicals and ozone) it is very likely that these compounds getting lost during analysis, i.e. under field conditions when air composition cannot be controlled accurately as in the laboratory. The present manuscript describes laboratory experiments with tropical soils from Amazonia and surrounding areas demonstrating SQT emission. By controlled drying out cycles under different atmospheric conditions (anaerob and aerob) the SQT emissions were parameterized and used to develop an emission algorithm enabling the calculation of SQT fluxes from the selected tropical soil types. Based on this algorithm a model exercise was performed comparing the SQT fluxes from canopy and soils at the ATTO site in Brazil.

The manuscript is very interesting describing novel findings. Overall the work shows that SQT emissions from tropical soil are highly variable depending on the soil type, with highest emission rates found in soil samples from "Tera firme" forest. It's also interesting to note that these emissions originate from bacteria but not from fungi (at least in the measured soils) indicating different microbial activities. This is also obvious for the large variation of SQT emission from the TF samples 1-3. Despite their high degree of similarity between physiochemical properties the emission rates differed strongly.

Reviewer#3 comment: This brings me to the critical points of the manuscript. The data are novel in the sense that they describe soil SQT emission from a globally important ecosystem. However, I have doubt that based on the current data a solid soil emission model can be drawn. The model was fit

with the current data and also validation was done with the same samples. Therefore, it is not astonishing that the experimental data and modeling output fit. As seen in Fig 3c, the overlay of field data and the modeled data is not as clear. The comparison of modeled canopy and soil SQT fluxes shows a potentially very high contribution of SQTs from the soil to the overall SQT flux. Keeping in mind the soil surface area relative to the much larger leaf surface per ground area in the canopy (MODIS LAI for Amazonian rain forest from c. 4-6 m² m⁻²; Doughty and Goulden, 2008 JGR) the SQT emissions from tropical soils are very high. This has to be brought in context with the few published reports on soil SQT emissions. Overall, the discussion is very weak (condensed), focusing on air chemistry rather on the possible biology behind the SQT emissions from soil.

Beside this shortcoming I see some additional points, which should be improved in the different sections of the manuscript i.e. in the presentation of the data.

Response by authors: We thank the reviewer for recognizing the novelty of our findings and highlighting the flux rate-surface area point. However, from the reviewer's comment "*it is not astonishing that the experimental data and modeling output fit*", it appears that that reviewer did not appreciate that the model is based on lab data while the validation is done with field data. "*As seen in Fig 3c, the overlay of field data and the modeled data is not as clear.*" We were in fact very pleasantly surprised by the fact that our model can describe so well the processes in a soil sample. We kindly refer the reviewer to the light and temperature dependency of isoprene and monoterpenes in several previous studies and for example the breakthrough paper of Guenther et al., 1993 on which most emission models are still based. One can see that while these empirical trends can be captured, there is always a large variation between measurements and model. Therefore, we do believe that our field measurements are captured very well by our algorithm.

In addition to the above, we would like to emphasize it is not self-evident that such lab experiments can be directly translated into field conditions. The good agreement we achieve in this study suggests that the main controlling processes have been taken into account.

Discussion comment: We agree with the reviewer that a section discussion should be included and critically discuss the context of the study. Given the generous space allocation offered by Nature Communications, we now include a new discussion section in which we address the results of the study in detail.

Methods

Reviewer#3 comment: For the reader it is very difficult to follow the numbers S1 to S? On line 402 it is stated that 38 samples were measured. In maximum I have seen data up to S25 (Extended data figure 2. So where is the rest? You should follow a clear layout e.g. according to the soil samples and soil layers.

Response by authors: We kindly refer the reviewer to our overview table (Ext.data table 1) where all samples are included and a clear layout is followed. In this table N=42, which is the sum of 38 samples with EXPSET1 and 4 samples with EXPSET2. This total number of samples measured was already included in the main text (L51). In addition, Ext.data Fig. 2 illustrate the emissions measured for the organic (O) horizon, as indicated in the figure's caption.

In order to avoid misunderstanding in the sample statistics the following revisions in the method's text have been made:

"A dynamic laboratory incubation system allowing automated measurement of 38 samples was used to investigate the release and uptake rates of VOCs from soil. In addition, 4 samples were measured with a slightly different setup that it is described below (see Ext data Table 1 for an overview). The chamber system that was used for the majority of the samples is described in detail elsewhere. Briefly,"

Reviewer#3 comment: Line 401ff: The cuvette system uses an active flow through the soil sample. That's not the case in nature. Under natural conditions movement of gaseous compounds are driven in soil by diffusion and not an active flux. How do you think your experimental set up has influenced the SQT emissions rates? It might be that the emissions are high because of this active gas stream. Please discuss this point with respect to your field system and published data where the incoming cuvette air streams over the soil and not through the soil pores.

Response by authors: The reviewer makes a good point that has been considered previously by soil scientists using such systems. In essence it is suggested that some sort compensation point may be present for the soils, in which case applying an active flow would increase the measured emission flux. We add this caveat to the main text for completeness, however we would like to point out that this is a well-established technique for measuring soil emissions that has shown good agreement between lab and field studies for nitric oxide (NO)(Behrendt et al., BG, 2014; Oswald et al., Science, 2013).

Addition in the methods section (lab experiments):

"It normally took 4-7 days to dry the soil completely, allowing us to investigate fluxes over the drying out period. While such conditions may not represent the natural diffusion driven exchange, it has been shown that our flow-through experimental setup can reliably quantify gaseous emissions from soil, such as NO (Behrendt et al., 2014; Oswald et al., 2013; Meixner et al., 2006)"

Addition in the methods (field measurements):

"At the ATTO site synthetic air from pressurized gas bottle was used for the ingoing air. Applying an active flux though the chamber may result in an overestimation of the quantified VOCs but it will not affect the emission pattern of sampled species."

Reviewer#3 comment: Line 462: E/N is changing strongly when drift tube pressure changes from 2 to 2.2. mbar. Please give the range. As mentioned in line 477 the fragmentation pattern strongly depends on E/N. Hence it is important how strongly your E/N differs.

Response by authors: This is correct but the range included refers to all experiments, conducted in different periods and it was always chosen to be set as such so the E/N ratio is 117 Td. That ensured that the fragmentation was always constant. As shown in Kim et al. (AMT, 2009), E/N ratio has indeed dramatic impacts on the fragmentation patters but only for dramatic changes in the ratio. Therefore $\pm 1-3\%$ of E/N would have not impacted the fragmentation pattern significantly.

Reviewer#3 comment: line 564 and related Fig3. How have you measured soil emission fluxes when a rain event occurred? Have you removed you soil chambers during the rain fall and put them back later?

Response by authors: Added as *"The experiments were set up directly after the rain event and hence the soil was not covered when rainfall occurred."*

Reviewer#3 comment: Figures are highly complex and in some cases hard to decipher:

Figure 1: Overall, this figure is rather complex. It would benefit from a re-organization e.g. SQT pattern and SQT legends separately from the curves. Don't see the need of the background image. In (a) what does the blue line represents? I assume you show monoterpenes. What does the shaded area in (a) represents? In the legend in Fig 1e the asterisk indicates that the subsoil was measured under aerobic conditions. Why? Why now showing the regular anaerobic situation? How many n are measured?

Response by authors: All the comments are now addressed in the figures legend. For the purposes of this response:

- The blue line was acetic acid. This is now removed for clarity.

- The shaded areas represent the standard deviation of an averaged measurement point (last 4 PTRMS points before chamber switch). More details are now provided in the methods section.
- The purpose of the figure was to give an example vertical profile of SQT emissions from each soil layer, and to show what is released into the atmosphere according to the model. Subsoil (B) is usually anaerobic but due to experimental limitations (limited N₂ bottles) the dilution gas was cleaned air produced by our zero air generator. This information is now included.

- Fig1. *“The shaded areas indicate the standard deviation of the measured emissions rates at each chamber cycle”*

-Fig1. *“The asterisk in panel e denotes that the experiments were performed under aerobic conditions despite the commonly dominating anaerobic conditions in the deep soil layers”*

While we agree that the background does not serve a scientific purpose, we would suggest the simplest form that is depicted below as it still serves the purpose of showing the vertical profile within and above the soil layers. In addition, we provide the simple form suggested by the reviewer. If the editor considers it as better option, we can certainly include the figure without background.

Suggested revision of Fig 1 by authors (left) and suggested revision by reviewer#3 (right)

Reviewer#3 comment: Figure 2:

Legend: which soil samples were shown O, A or B? Please indicate.

Response by authors: It is now indicated as suggested:

“SQT emissions (markers, dashed line as model output) and 16S- / 18S- rRNA transcript abundance (TA) during desiccation of soil samples from organic (O) horizon from TF4 and TF5.”

Reviewer#3 comment: Figure 3 and general comment:

It is not astonishing that modeled curves fit to experimental data when the later are used as input parameters for the 2 submodels (HM and MM). How does the general model perform for the other soil samples? Have you tested / validated your algorithm with other samples not used for parameterization?

Response by authors: As we note in L132 “Close agreement between the laboratory measurements and the emission model has been achieved for all experiments with optimum emission functions ($0.89 < R^2 < 0.97$)”. This is a point that has been raised also by reviewer#2 and shows that this misconception can arise from this condensed description. We now make it clearer that our model can effectively capture all experiments that displayed emissions under optimum (MM).

“The emission model algorithm was applied in all laboratory experiments where an optimum emission at a certain WFPS was observed (i.e. organic (O) horizon topsoil (A)). A close agreement between the emission algorithm and measured emissions ($0.89 < R^2 < 0.97$) indicates that the algorithm can reliably simulate the observed SQT emissions (see examples in Fig. 3a and 3b)”

Reviewer#3 comment: Fig 3a: the grey symbols mean aerobic and not anaerobic.

Response by authors: We thank the reviewer for noticing this mistake. This is now corrected.

Reviewer#3 comment: Fig 3b: no SQT measurements under anaerobic are given for the topsoil (a). Does it mean that in this layer at aWFPS of e.g. 90% the conditions are aerobic while in the O horizon at the same WFPS the conditions are anaerobic?

Response by authors: This is a fair point. However, our hydrological model indicates that saturation is much more frequent in the organic (O) horizon, even if it certainly occurs for the topsoil (A) .

We clarify this in the main text as following:

“Our field measurements quantified exceptionally strong emissions of SQT, 6 hours after strong rainfall (25.1mm in 2 hours). These emissions were stronger than our model prediction indicating that either the emission burst could be stronger compared with the laboratory observations or that the topsoil (A) significantly contributes to anaerobic emission burst. We note that according to our hydrological model, topsoil (A) is very rarely under anaerobic conditions and hence such conditions were not included in the emission model. The chemical speciation of the first...”

Fig 3c: what does the left part with the box ? Was this TF1 sample measured a day 2? Remove for clarity.

Response by authors: The reviewer interpreted the figure correctly. TF1 is located in a remote area and hence any sampling there is challenging. What we wanted to show in this picture is the fact that field conditions compared well with laboratory simulations in both soils and in the order of magnitude emission strength. Following the suggestion by the reviewer, the TF1 is now removed for clarity and the observation is described in the text:

“In addition to TF3, field measurements were conducted at the strong emitter soil TF1. Similar to TF3, the emission strength was predicted by our model algorithm (model prediction: $114 \mu\text{g m}^{-2} \text{h}^{-1}$, field measurements $100 \pm 56 \mu\text{g m}^{-2} \text{h}^{-1}$).”

Reviewer#3 comment: Figure 4:

Legend: “Daily averages of modeled SQT emissions....” How does the data look like for TF2? What’s with this sample? It’s barely mentioned in the paper (except extended data figures 1 and 2). Why haven’t you used all samples from the ATTO side to show the high variability of soil SQT emissions.

Response by authors: The reason behind our limited attention to TF2 is the following: TF1 displayed extraordinary high emissions and therefore most of the attention was given in this site. Similar attention was given to TF3 which is actually where the ATTO station is located and all continuous

measurements are running. TF2 was located somewhere in between displaying similar emissions to TF3 and hence our discussions focus only in the variability. Nonetheless, we now include TF2 in the speciation Table 2 in addition to the already presented data of Ext. Figs 2 and 3.

We believe that the high variability is properly addressed by Ext.data Fig2. We now bring forward this figure and extensively discuss the variability and values observed in the discussion section.

Reviewer#3 comment: Extended data figure 1:

How you define outliers? It seems they only occurred in the 0-5 cm horizon O.

At which WFPS you consider the samples to be anaerobic? Please define.

Response by authors: *"The boxplot draws points as outliers if they are greater than $q3 + w \times (q3 - q1)$ or less than $q1 - w \times (q3 - q1)$. $q1$ and $q3$ are the 25th and 75th percentiles of the sample data, respectively. The default value for 'Whisker' corresponds to approximately $\pm 1.5 \times (q3 - q1)$ and 99.3 percent coverage if the data are normally distributed. The plotted whisker extends to the adjacent value, which is the most extreme data value that is not an outlier.*

(<https://de.mathworks.com/help/stats/box-plots.html>)".

The correct observation by the reviewer has do with the exceptional high emissions observed at the very first samples of some O-horizon soil experiments.

Anaerobic conditions were considered to occur at the beginning of the wetting stage when WFPS was >80% (L128-129). Considering the general experimental approach, anaerobic samples were conducted with the use of a pressurized bottle of nitrogen (L61, L279). Nonetheless, the cross-regime WFPS is based on an assumption that was built by the patterns observed in the laboratory and in the field.

Reviewer#3 comment: Extended data figure 2:

Please indicate /mark which sample number belongs to which soil sample. Here you mention for some samples WFPSs of c. 45 % defining them as anaerobic.

Response by authors: The numbers on x-axis indicate the respective sample number of Table 1. This information is now added in the Table's caption.

On L334, we define anaerobic the experiments that have been conducted in the absence of oxygen (using N₂ as dilution gas for the main air stream).

Reviewer#3 comment: In Figure 3a you classified anaerobic conditions down to a WFPS of c. 90% in the O layer. In Fig 3b no anaerobic conditions at all, though the WFPS was above 90%. So what's true? At which WFPS you have a change from aerobic to anaerobic conditions? Please be consistent throughout the manuscript.

Response by authors: We kindly refer the reviewer to our previous response on Fig. 3b (5 comments above). In addition, we note that for the experiments that used synthetic air as dilution, anaerobic conditions were considered only when WFPS was >80% while for the experiments that we used anaerobic conditions (achieved by oxygen free conditions and the use of pure nitrogen). Since there was a constant flow of air containing oxygen, we used the initial HM part from the N₂ experiments (organic (O) horizon only). Given the hydrological model indications, the magnitude and the time required for SQTs to be released in the atmosphere, we did not consider it necessary to include the same approach on the (A) horizon. This is now stated in the text (as our response above).

Reviewer#3 comment: Extended data figure 3:

Which soil layers are used for this comparison? In(c) you show much more data points as in (a) and (b). Where do they come from? SQT emissions from PTR-MS data were calculated from m205. Please indicate.

Response by authors: In (c) we used all laboratory data available and from all soil horizons. The summed N = 27 as indicated in the figure. This number is a product of the 25 laboratory comparisons (indicated in table 2) plus 2 samples from TF2 that was not included in the table as we don't have field measurements. In (a) and (b) we simply illustrate an exemplary comparison during the two separate desiccation experiments. In order to make it clearer, we now include TF2 in the table and the text has been modified describing in more detail the presented data. Also please note that Ext.data Fig. 3 is now revised in order to additionally present the linear intercomparison between PTR-MS and GC-MS measurements as requested by reviewer#1.

Final remark by authors to Reviewer#3: We highly appreciate all the comments made. Your review indicates several misconceptions that may arise from the initial version of the manuscript. We hope that we have sufficiently revised our text and figures and that now our experimental approach, definitions and discussions are comprehensive.

Results section added:

Forest canopy model. *The average dry season SQT emissions from TF soils ($44.9 \mu\text{g m}^{-2} \text{h}^{-1}$ considering emissions from TF1, TF2, TF3 and TF4) were incorporated into a simple forest canopy column model³³ (FORCAsT (FORest Canopy Atmosphere Transfer)) in order to evaluate the significance of soil SQTs to air chemistry. The majority of these highly reactive species were lost through ozonolysis within the canopy space, with a maximum of 1.5% of soil emissions escaping the canopy just before dawn when photochemistry commences and around 0.2% through most of the day as reaction rates reach a maximum. Soil emissions of SQTs account for as much as 50% of O₃ reactivity (soon after dawn) at the soil surface and a relatively constant 30% during daylight hours. Overnight, soil SQTs contribute nearly 40% of O₃ reactivity at the top of the canopy, but this is substantially reduced to 0.5-1% during daylight hours as the SQTs are rapidly consumed within the canopy. The high reactivity of SQTs results in substantial enhancements in the concentrations of condensable reaction products that will partition into the aerosol phase and potentially act as CCN. SQT-derived oxidation products increase the total concentration of condensable species at the top of the canopy by 30-40% at night when reaction rates are low for other emitted compounds decreasing to around 20% at dusk when the contribution from other species is highest.*

Discussion section added:

Discussion. *Strong sesquiterpene (SQT) emissions from Amazonian soils have been quantified from laboratory investigations. A reproducible emission pattern as a function of water filled pore space (WFPS) was recognized and an empirical emission algorithm was created assuming different microbial processes under two different moisture regimes (high moisture/initial burst HM and medium moisture/drying out MM).*

The initial emission burst of SQTs observed at the beginning of each experiment (HM regime) have been variously ascribed to i) hypo-osmotic stress response of the soil microbes, ii) the replacement of head space air by water and its release back to the atmosphere and iii) activation of rapid, intermediate and/or delayed responding soil microbes. Chemical speciation of SQT emission rates obtained in the field during September 2016 (TF3), showed distinctly different SQT species were emitted 6h after a strong rainfall event compared to the following dry days showing that different (and possibly multiple) processes are responsible for the SQT emission bursts following rainfall.

Under the MM regime, SQTs displayed optimum emissions at certain WFPS reproducibly, thus indicating the most favorable conditions of microbially produced SQTs. The monitoring of 16S- and 18S-rRNA transcripts was undertaken as a practical means to study microbial community responses to environmental variables. While some taxa do not demonstrate a linear correlation between rRNA abundance and microbial activity, ribosomes indicating potential cell activity in a community are more abundant than in dormant cells. At the community level, an increase in rRNA content is assumed to reflect greater protein production over time, and therefore a proxy for activity. Our experiments demonstrated a variation of transcript abundance between two ecologically diverse soils, and in addition, they showed a peak of transcription coincident with SQT peak emissions in the MM regime. This strongly suggests that microbial activity inferred from ribosomal can be associated with SQT production and release.

Emission rates from two TF soils, simulated over the course of 2 years, predicted stronger emissions during the dry season when prevailing conditions led to the WFPS associated with optimum emissions. During the El Niño year of 2015, extended drought conditions led to a slight decrease of SQT emission rates, as the field WFPS dropped to about 25-30 % for the organic (O) horizon. Nonetheless, this falls inside the uncertainty limits for the optimum WFPS observed in the laboratory and the direct implications of extended drought to microbial activity and subsequent SQT release requires further investigation. The emission strength however, was validated with field measurements for both TF3 (ATTO site) and the strongest emitter TF1. The comparison with an established emission algorithm for vegetation (MEGAN) indicated that soil SQT emissions from TF1 can rival the canopy emissions at the same location, being of comparable magnitude in the dry season. The inherent uncertainties included in this comparison primarily originate from the temperature dependency (β -factor) that is used for vegetation emissions, and the large variability that has been observed for the soil emissions. To the best of our knowledge, no studies have directly addressed and determined the temperature dependency of SQT emissions from Amazonian vegetation. Hence, the present implicit assumption of a common and stable throughout a season β -factor =0.17 (as used in MEGAN for all SQT species) can be considered a rough approximation with large associated uncertainty. It has been shown that the β -factor varies significantly day-to-day, and with atmospheric conditions⁹. In addition to canopy model uncertainties, soil emissions displayed significant variation in strength between the sites investigated, with emission estimates ranging from a few $\mu\text{g m}^{-2} \text{h}^{-1}$ to orders of magnitude higher. In general soil emissions seem to be primarily location and microbial activity specific, as demonstrated by TF4 and TF5.

The average dry season SQT emissions from Terra Firme soils were incorporated into a simple forest canopy column model³³. The results indicated that soil SQTs dominate the ozone reactivity close to the forest floor. Soil emissions from TF1 alone, could account as much as 75% of O₃ reactivity (soon after dawn) close to the soil surface and have a relatively constant contribution of 50% during daytime. While a small fraction of these emissions can escape the canopy, their oxidation products will be considerable on the canopy top. The model thus suggest that not only do emissions of SQTs from Terra Firme soils substantially affect atmospheric chemistry within the canopy but that the effects have the potential to alter regional chemistry-climate.

Reviewers' comments:

Reviewer #1 (Remarks to the Author):

I have read with much pleasure the new paper version and the letter response. The authors readily improved the manuscript and well defended their experimental work in response to my critical remarks. The presentation and handling of data is now clearer and the inclusion of the canopy chemistry box model indeed elucidated the overall conclusions. I also appreciated the way the authors discuss unknown variables and (unavoidable) limitations of their study.

I offer here a few remarks the authors may consider for further improvement:

1) Despite the authors response, the fumigation of soils with VOC in the laboratory experiments remains somewhat elusive. The statement "14 VOC components... introduced in atmospherically relevant VOC mixing ratios (0.1-1ppb)" is quite vague. The authors should better explain the idea behind this fumigation and according to which criteria the authors have chosen the VOC species and the concentrations they applied? Following the authors comment soils were fumigated at least with methanol and some monoterpenes, for which mostly an uptake was observed. So what about fumigation with SQTs, for which no or little uptake but consistent emissions were observed? I guess there was no SQT fumigation, because these VOC are quite sticky and reactive making it difficult to generate reliable gas standards in tanks. From a theoretical point of view, VOC fumigation is desired when the concentration of the regarded VOC in the headspace is largely below the concentrations occurring under natural conditions thus creating unnatural strong VOC gradients that could possibly boost the net emission rate. Here the question arises what are the natural VOC concentrations above isolated soil horizons? For the upper organic horizon the authors may rely on field ambient air concentrations measured close to the ground, which for SQTs were probably rather low. But what about the lower horizons that were all measured separately in the laboratory experiments? Given that the upper organic horizon emits quite high amounts of SQTs, it is likely that high concentrations of SQTs are reached inside its air spaces, which could negatively feedback on the net SQT exchange rate of the soil horizons below.

In this context, the authors make two interesting statements in the section "METHODS". First L 579-582: "In EXPSET2 we did not fumigate with environmentally relevant VOCs therefore..", further line 676-679: "Assuming a compensation point between ambient air and within soil SQT concentrations, the use of zero air...". These are important remarks but somewhat dispersed and drown under in the manuscript. It would be valuable for the paper to precise more the concept and methods of VOC fumigation and discuss consistently all aspects and questions arising with this topic somewhere in the section DISCUSSION for example when comparing results from laboratory experiments with those from field measurements. In the same context it would be interesting to know how the exchange of VOC other than SQT behaved in the field (compared to the lab experiments). For

example were monoterpenes emitted from soils in the field or mainly taken up as observed in the lab?

Another issue that could be briefly addressed in the discussion when comparing field with laboratory results is the possible role of an intact food web on soil VOC emission. Unless there were some compensatory “artefacts” on measured VOC emission rates in field and lab (see above for example), the total lack of divergence between soil emissions from laboratory and field experiments would mean that the presence of other soil organisms including, fungi, roots macro and micro fauna has no effect on net VOC emissions from soils in neotropical forests. This is pretty unlikely and to some extent contradicts many ecological studies that showed various VOC mediated interactions among below ground organisms in the same or other ecosystems. The presence/absence of other organisms and their interactions with soil bacteria might be the likely reason that laboratory measurements (and hence model predictions) at least occasionally diverged from field observations, a point which could be more underlined in the Discussion.

Finally the overall outcome of the study suggests that soil bacteria in the Amazonian basin release mainly SQTs to the air. This is somewhat surprising, because according to the rapidly expanding literature, bacteria and soil organisms usually produce a wealth of other VOC such as oxygenated alkenes and alkanes and other non-terpenoid VOCs (see e.g. recent review by Piechulla and collaborators (PCE 2017) and mVOC database <https://doi.org/10.1093/nar/gkx1016>). Perhaps the authors observed peaks/masses of such VOC but decided to focus on SQTs and few other VOC they explicitly could identify. If this is the case, it should be briefly mentioned in the manuscript.

Few minor comments

L. 114 insert full stop

L122 “After the rain, the SQT emission burst declines exponentially and then stabilizes as the optimum emission is reached.” Check meaning; “decline” versus “optimum”?

L179 <CCN> remove abbreviation if not used further in the text.

L183 and L185: No need to reintroduce the abbreviations <SQT> and <WFPS>.

L274: Figure 1 caption: (please note which of the VOC fluxes obtained from laboratory experiments included fumigation of the same VOC.

L357:< The green bullets and bars in c indicate the std of...> I guess they indicate <means and std...>.

I congratulate all authors on this excellent study and thank for their insightful comments on my remarks.

Reviewer #4 (Remarks to the Author):

The current manuscript addresses an important and actual topic: significant source of highly reactive VOCs (sesquiterpenes) from forest soil that affects air chemistry and potentially, influence climate. The topic is of global biogeochemistry importance, and the findings are novel. The strength of the experimental design is the location of the field experiment, at the Amazon rainforest, which is for sure an excellent and relevant location to study. The experiments are well-performed and the manuscript is well-written and concise. The current revised manuscript will find a broad interest among the readers of nature communications. Overall, the revised version of this manuscript has addressed all the main concerns raised previously by the reviewers.

The weakest point of the manuscript remains their emission model, based on laboratory experiments only, and using soil without plants. Plant-microbe interactions are fundamental in soil, so it is unclear how the authors can develop a reliable VOC model based on soil without rhizosphere, which is the most active fraction of the soil. The microbiome of a soil without plants is completely different than the microbiome of soil containing plants.

Another critical point is the validation of the model. This was performed using very few field measurements, specifically 8 measurement time-points spanning for only 4 days. Although the authors were able to fairly-well catch the soil-forest SQT emissions and therefore validate the relevance of their study, it is unclear how such model will perform using data collected throughout the whole year. Seasonal variations of SQT emissions might be expected by seasonal changes of soil composition and microbiome community. Another point is the lack of a proper sensitivity study to address uncertainties of input parameters affecting the results.

I am aware that my points raised above are outside of the main aim of the present manuscript and that the actual rudimentary modeling approach might be acceptable for the current goal of the paper, i.e. demonstrating a significant and relevant source of SQT outside the green-tissues of forests.

Nevertheless, the authors should make clear in the manuscript the limit of their modeling approach, and state that a proper modeling approach is required in the next future. This will avoid concerns and misinterpretation of the results among readers.

Minor points:

Sesquiterpene emissions from fungi are strongly dependent on fungal-age, rather than biomass (Weigl et al., 2016). Fungal sesquiterpene emissions are also exceptionally strong during their initial growing phase (Weigl et al., 2016). I don't think that the authors can conclude that the fungal emission was insignificant (L101-103). This would require further support from measurements of individually measured fungi and bacteria to confirm author's statement. Actually, current knowledge points to fungi as a potential strong source of SQT (e.g. Schnürer et al., 1999; Caruthers et al., 2000; Rösecke et al., 2000; Ziegenbein et al., 2006; Hynes et al., 2007; Effmert et al., 2012; Polizzi et al., 2012a; Polizzi et al., 2012b; Crutcher et al., 2013; Heddergott et al., 2014; Weigl et al., 2016).

Among all literature, the paper of Weigl et al., 2016 is highly important, since both fungi *Alternaria alternata* and *Fusarium oxysporum* are i) strong sesquiterpene emitters ii) saprotrophic and iii) highly ubiquitous fungi. Therefore, those fungi are everywhere in forests and likely constitute a potential and significant source of sesquiterpenes.

L50: 'soil volatilomics'. This term may sound 'sexy' to someone, but it disturbs others (including me). What is 'soil volatilomics' for the authors? Have the authors been measuring inorganic volatiles? I can't find any data on methane, nitrous oxides, ammonia, dihydrogen sulphide...

The fitness of the model was calculated based on the same sample set (laboratory data) (L131-132). This is unfortunate. Cross-validation should be performed on two independent set of data, the first is used for training the model, the second for evaluating the model.

From Fig.5 it is quantitatively unclear how modeled and measured field data are matching to each other.

References:

Caruthers JM, Kang I, Rynkiewicz MJ, Cane DE, Christianson DW (2000) Crystal structure determination of aristolochene synthase from the blue cheese mold, *Penicillium roqueforti*. *J Biol Chem* 275: 25533–25539

Crutcher FK, Parich A, Schuhmacher R, Mukherjee PK, Zeilinger S, Kenerley CM (2013) A putative terpene cyclase, *vir4*, is responsible for the biosynthesis of volatile terpene compounds in the biocontrol fungus *Trichoderma virens*. *Fungal Genet Biol* 56: 67–77

Effmert U, Kalderás J, Warnke R, Piechulla B (2012) Volatile mediated interactions between bacteria and fungi in the soil. *J Chem Ecol* 38: 665–703

Heddergott C, Latgé JP, Calvo AM (2014) The volatome of *Aspergillus fumigatus*. *Eukaryot Cell* 13: 1014–1025

Hynes J, Müller CT, Jones TH, Boddy L (2007) Changes in volatile production during the course of fungal mycelial interactions between *Hypholoma fasciculare* and *Resinicium bicolor*. *J Chem Ecol* 33: 43–57

Polizzi V, Adams A, Malysheva S V., De Saeger S, Van Peteghem C, Moretti A, Picco AM, De Kimpe N (2012a) Identification of volatile markers for indoor fungal growth and chemotaxonomic classification of *Aspergillus* species. *Fungal Biol* 116: 941–953

Polizzi V, Adams A, De Saeger S, Van Peteghem C, Moretti A, De Kimpe N (2012b) Influence of various growth parameters on fungal growth and volatile metabolite production by indoor molds. *Sci Total Environ* 414: 277–286

Rösecke J, Pietsch M, König WA (2000) Volatile constituents of wood-rotting basidiomycetes. *Phytochemistry* 54: 747–750

Schnürer J, Olsson J, Börjesson T (1999) Fungal volatiles as indicators of food and feeds spoilage. *Fungal Genet Biol* 27: 209–217

Weigl F, Ghirardo A, Schnitzler J-P, Pritsch K (2016) Sesquiterpene emissions from *Alternaria alternata* and *Fusarium oxysporum*: Effects of age, nutrient availability, and co-cultivation. *Sci Rep* 6: 22152

Ziegenbein FC, Hanssen HP, König WA (2006) Secondary metabolites from *Ganoderma lucidum* and *Spongiporus leucomallellus*. *Phytochemistry* 67: 202–211

Reviewers' comments:

Reviewer #1 (Remarks to the Author):

I have read with much pleasure the new paper version and the letter response. The authors readily improved the manuscript and well defended their experimental work in response to my critical remarks. The presentation and handling of data is now clearer and the inclusion of the canopy chemistry box model indeed elucidated the overall conclusions. I also appreciated the way the authors discuss unknown variables and (unavoidable) limitations of their study.

We thank the reviewer for the insightful feedback and for acknowledging the improvements in the revised version. It is evident that the comments received helped us to restructure the manuscript so the results clearly illustrate the findings and the unknowns are properly discussed. We further appreciate the additional remarks below that are now also addressed in the final version.

I offer here a few remarks the authors may consider for further improvement:

1) Despite the authors response, the fumigation of soils with VOC in the laboratory experiments remains somewhat elusive. The statement "14 VOC components... introduced in atmospherically relevant VOC mixing ratios (0.1-1ppb)" is quite vague. The authors should better explain the idea behind this fumigation and according to which criteria the authors have chosen the VOC species and the concentrations they applied? Following the authors comment soils were fumigated at least with methanol and some monoterpenes, for which mostly an uptake was observed. So what about fumigation with SQTs, for which no or little uptake but consistent emissions were observed? I guess there was no SQT fumigation, because these VOC are quite sticky and reactive making it difficult to generate reliable gas standards in tanks. From a theoretical point of view, VOC fumigation is desired when the concentration of the regarded VOC in the headspace is largely below the concentrations occurring under natural conditions thus creating unnatural strong VOC gradients that could possibly boost the net emission rate. Here the question arises what are the natural VOC concentrations above isolated soil horizons? For the upper organic horizon the authors may rely on field ambient air concentrations measured close to the ground, which for SQTs were probably rather low. But what about the lower horizons that were all measured separately in the laboratory experiments? Given that the upper organic horizon emits quite high amounts of SQTs, it is likely that high concentrations of SQTs are reached inside its air spaces, which could negatively feedback on the net SQT exchange rate of the soil horizons below.

In this context, the authors make two interesting statements in the section "METHODS". First L 579-582: "In EXPSET2 we did not fumigate with environmentally relevant VOCs therefore..", further line 676-679: "Assuming a compensation point between ambient air and within soil SQT concentrations, the use of zero air...". These are important remarks but somewhat dispersed and drown under in the manuscript. It would be valuable for the paper to precise more the concept and methods of VOC fumigation and discuss consistently all aspects and questions arising with this topic somewhere in the section DISCUSSION for example when comparing results from laboratory experiments with those from field measurements. In the same context it would be interesting to know how the exchange of VOC other than SQT behaved in the field (compared to the lab experiments). For example were monoterpenes emitted from soils in the field or mainly taken up as observed in the lab?

The reviewer correctly points out the possible limitations that may arise from fumigating the soils with VOC and correctly surmises that we did not fumigate with sesquiterpenes as pressurized gas

standards are not available for these species. The criteria used were the ambient mixing ratios close to the forest floor, as they have been previously measured with in-situ campaigns, in addition to the investigation of uptake gradients. As commonly agreed, this is the best approach for the top soil (Organic (O) horizon) and could potentially feedback on the net SQT exchange from the middle soil horizon. Nonetheless, it is evident that the emissions from the top soil are markedly stronger from the Organic (O) horizon and in addition, our field measurements did not reveal a discrepancy that would have implied an important role of such dynamics. Therefore, we can conclude that in addition to experimental difficulties, a possible negative feedback on the net exchange from the low emitter soil layers is negligible. We agree that this point should be included and therefore we have added additional text in the discussion section as suggested.

Regarding the reviewers interest in further species, we are working on a separate publication detailing the behaviour of the other VOCs measured. The sesquiterpene story as described here, struck us as the most interesting and so we have examined it first. To answer the reviewer's question, however, monoterpenes are also weakly emitted in the relevant WFPS and are strongly uptaken only under very dry conditions.

Please see the two new discussion paragraphs provided at the end of this response.

Another issue that could be briefly addressed in the discussion when comparing field with laboratory results is the possible role of an intact food web on soil VOC emission. Unless there were some compensatory "artefacts" on measured VOC emission rates in field and lab (see above for example), the total lack of divergence between soil emissions from laboratory and field experiments would mean that the presence of other soil organisms including, fungi, roots macro and micro fauna has no effect on net VOC emissions from soils in neotropical forests. This is pretty unlikely and to some extent contradicts many ecological studies that showed various VOC mediated interactions among below ground organisms in the same or other ecosystems. The presence/absence of other organisms and their interactions with soil bacteria might be the likely reason that laboratory measurements (and hence model predictions) at least occasionally diverged from field observations, a point which could be more underlined in the Discussion.

This is a good point to emphasize when dealing with a subsection of an ecological system. The soil does interact with organisms, fungi, roots and this should be kept in mind when assessing our experimental determination of the soil emission rates. This point, which was also raised by reviewer 4, is taken up in the discussion section.

Please see the new discussion paragraphs provided at the end of this response.

Finally the overall outcome of the study suggests that soil bacteria in the Amazonian basin release mainly SQTs to the air. This is somewhat surprising, because according to the rapidly expanding literature, bacteria and soil organisms usually produce a wealth of other VOC such as oxygenated alkenes and alkanes and other non-terpenoid VOCs (see e.g. recent review by Piechulla and collaborators (PCE 2017) and mVOC database <https://doi.org/10.1093/nar/gkx1016>). Perhaps the authors observed peaks/masses of such VOC but decided to focus on SQTs and few other VOC they explicitly could identify. If this is the case, it should be briefly mentioned in the manuscript.

For our experiments, we have employed a quadrupole PTR-MS system that monitors a limited number of species. Alkanes and small alkenes cannot be detected due to their low proton affinity. The most important oxygenated species (acetone, methanol) were indeed monitored and included

in the manuscript (figure 1). In this study, we have focused on the highly reactive SQTs as they reveal a new important link between soil emissions and air chemistry. The behavior of the other VOCs will be reported in a separate publication.

Few minor comments

L. 114 insert full stop

L122 “After the rain, the SQT emission burst declines exponentially and then stabilizes as the optimum emission is reached.” Check meaning; “decline” versus “optimum”?

Changed to “After the rain, the SQT emission burst declines exponentially and then stabilizes as the optimum shaped microbial emissions start to increase”.

L179 <CCN> remove abbreviation if not used further in the text.

L183 and L185: No need to reintroduce the abbreviations <SQT> and <WFPS>.

L274: Figure 1 caption: (please note which of the VOC fluxes obtained from laboratory experiments included fumigation of the same VOC.

L357:< The green bullets and bars in c indicate the std of...> I guess they indicate <means and std...>.

I congratulate all authors on this excellent study and thank for their insightful comments on my remarks.

All minor comments have been addressed. We are grateful for the in-depth comments provided by the reviewer that has helped us to set our results in context.

Reviewer #4 (Remarks to the Author):

The current manuscript addresses an important and actual topic: significant source of highly reactive VOCs (sesquiterpenes) from forest soil that affects air chemistry and potentially, influence climate. The topic is of global biogeochemistry importance, and the findings are novel. The strength of the experimental design is the location of the field experiment, at the Amazon rainforest, which is for sure an excellent and relevant location to study. The experiments are well-performed and the manuscript is well-written and concise. The current revised manuscript will find a broad interest among the readers of nature communications. Overall, the revised version of this manuscript has addressed all the main concerns raised previously by the reviewers.

The weakest point of the manuscript remains their emission model, based on laboratory experiments only, and using soil without plants. Plant-microbe interactions are fundamental in soil, so it is unclear how the authors can develop a reliable VOC model based on soil without rhizosphere, which is the most active fraction of the soil. The microbiome of a soil without plants is completely

different than the microbiome of soil containing plants.

Another critical point is the validation of the model. This was performed using very few field measurements, specifically 8 measurement time-points spanning for only 4 days. Although the authors were able to fairly-well catch the soil-forest SQT emissions and therefore validate the relevance of their study, it is unclear how such model will perform using data collected throughout the whole year. Seasonal variations of SQT emissions might be expected by seasonal changes of soil composition and microbiome community. Another point is the lack of a proper sensitivity study to address uncertainties of input parameters affecting the results.

I am aware that my points raised above are outside of the main aim of the present manuscript and that the actual rudimentary modeling approach might be acceptable for the current goal of the paper, i.e. demonstrating a significant and relevant source of SQT outside the green-tissues of forests.

Nevertheless, the authors should make clear in the manuscript the limit of their modeling approach, and state that a proper modeling approach is required in the next future. This will avoid concerns and misinterpretation of the results among readers.

We thank the reviewer for recognizing that the points raised above are outside the main aim of the present manuscript and for acknowledging the overall novelty of the experiments and modelling.

The reviewer notes correctly that a soil including a functioning rhizosphere may give different emissions to those we report. We argue that measuring the soil in isolation is of value as we may see specifically the potential emissions from this source and their strength. Here the interesting point is the strong source of SQTs, which can enter the atmosphere and impact atmospheric chemistry. We agree with the reviewer that our finding should be set in context and the limitations of measuring a subsection of the ecosystem noted in the text. We now do this in the discussion section. The reviewer also mentions that longer in-situ measurements would be desirable. We agree. The in-situ data we report here serve to validate the overall concept and longer term measurements in the future will test our new model.

Hence, we now additionally discuss them in the context that has been suggested by the reviewer. Please see the new discussion paragraphs provided at the end of this response.

Minor points:

Sesquiterpene emissions from fungi are strongly dependent on fungal-age, rather than biomass (Weigl et al., 2016). Fungal sesquiterpene emissions are also exceptionally strong during their initial growing phase (Weigl et al., 2016). I don't think that the authors can conclude that the fungal emission was insignificant (L101-103). This would require further support from measurements of individually measured fungi and bacteria to confirm author's statement. Actually, current knowledge points to fungi as a potential strong source of SQT (e.g. Schnürer et al., 1999; Caruthers et al., 2000; Rösecke et al., 2000; Ziegenbein et al., 2006; Hynes et al., 2007; Effmert et al., 2012; Polizzi et al., 2012a; Polizzi et al., 2012b; Crutcher et al., 2013; Heddergott et al., 2014; Weigl et al., 2016). Among all literature, the paper of Weigl et al., 2016 is highly important, since both fungi *Alternaria alternata* and *Fusarium oxysporum* are i) strong sesquiterpene emitters ii) saprotrophic and iii) highly ubiquitous fungi. Therefore, those fungi are everywhere in forests and likely constitute a potential and significant source of sesquiterpenes.

We appreciate the insightful feedback concerning fungal emissions. While we agree that further measurements are needed to confirm our observation (already noted in the text), we want to emphasize that we have carefully presented our measurements indicating that this is relevant for the particular ecosystem. To make it clearer, we have added the following sentence in the results section:

“TF4 and TF5 are typical for a drier region of the Amazonian basin and in addition, SQT emissions from fungi are strongly dependent on fungal-age, rather than biomass (Weigl et. al, 2016). Therefore the bacterial/fungal contribution to the net SQT production in the Amazon basin requires further investigation, particularly as a function of season and fungal development stage.”

In addition to this clarification, we have used the information provided, in the discussion of the strong discrepancy between the simulated emissions and the first point of field measurements. Please see the new discussion paragraphs provided at the end of this response.

L50: ‘soil volatilomics’. This term may sound ‘sexy’ to someone, but it disturbs others (including me). What is ‘soil volatilomics’ for the authors? Have the authors been measuring inorganic volatiles? I can’t find any data on methane, nitrous oxides, ammonia, dihydrogen sulphide...

Our intention was to report observations that fall inside the general research area of volatile organic compounds originating from soils hence “volatilomics”. It is a term in common use also in the emission of volatiles from people. However, so as not to confuse any readers this term has been now removed.

The fitness of the model was calculated based on the same sample set (laboratory data) (L131-132). This is unfortunate. Cross-validation should be performed on two independent set of data, the first is used for training the model, the second for evaluating the model.

From Fig.5 it is quantitatively unclear how modeled and measured field data are matching to each other.

Our intention in Fig. 5 was to illustrate how the emission algorithm was derived (Fig. 5a and 5b) and how the empirical parameterization obtained from laboratory samples can capture the emission dynamics observed in the field (Fig. 5c). Both simulated (thick red line) and in-situ measurements in the field (green circles) are quantitatively similar (emission rates are presented in the left y axis). In addition, we state in the text that *“Similar to TF3, the emission strength was predicted reasonably well by our model algorithm (model prediction: 114 $\mu\text{g m}^{-2}\text{h}^{-1}$, field measurements $100\pm 56 \mu\text{g m}^{-2}\text{h}^{-1}$).”*

The suggestions for additional discussion have been incorporated in the first part of the respective section as indicated with the green color.

“Discussion. *Strong sesquiterpene emissions from Amazonian soils have been quantified from laboratory investigations. A reproducible emission pattern as a function of WFPS was recognized and an empirical emission algorithm was created assuming different microbial processes under two different moisture regimes (high moisture/initial burst HM and medium moisture/drying out MM).*

The algorithm was developed based on desiccation experiments from sieved soils. While this approach allows the quantification of purely soil emitted VOCs, plant-microbe interactions and the possible role of an intact web on soil emissions is not addressed here. Other soil organisms (such as fungi, roots, micro and macro fauna) may influence the net effect of SQT emissions from soils in neotropical forests and it remains to be tested how the algorithm, evaluated with one in-situ measurement period, will perform under the seasonal changes of soil composition, microbiome community and plant developmental stages. Nonetheless, the emission pattern and strength forecasted at Fig. 5c indicates that laboratory incubations can be used as proxy for emissions in the field.

Our samples were fumigated with VOC mixing ratios at levels that have been previously measured in the field, although due to experimental limitations, SQTs were not introduced in the headspace of the chambers. Assuming a compensation point between ingoing air and within soil SQT concentrations, the creation of atypical concentration gradients could possibly increase the net emission rate. Apart from the physical processes, biological influences may add to the uncertainties. Shown in Fig. 4, the samples S13 and S14 of TF5 (also S16 and S17 of TF6) were analyzed with different experimental setups (Exp.Set.1 and Exp.Set.2 respectively (see methods for details)). The emission rates display the same behavior, and the emission rate differences, usually within the error bars, can be primarily attributed to inter-sample variability. Another possible reason could be the absence of VOCs in the main ingoing air stream of S13. In Exp.Set.2 we did not fumigate and therefore the lower production of SQTs observed (compared to S14; Exp.Set.1) could potentially indicate a connection between ambient-other than SQT- VOCs and microbial production in both regimes.

The initial emission burst of SQTs observed at the beginning of each experiment (HM regime) have been variously ascribed³⁴ to i) hypo-osmotic stress response of the soil microbes, ii) the replacement of head space air by water and its release back to the atmosphere and iii) activation of rapid, intermediate and/or delayed responding soil microbes³⁵. Chemical speciation of SQT emission rates obtained in the field during September 2016 (TF3), showed distinctly different SQT species were emitted 6h after a strong rainfall event compared to the following dry days showing that different (and possibly multiple) processes or sources are responsible for the SQT emission bursts following rainfall. Fungal emissions are exceptionally strong during their initial growing phase (Weigl et. al., 2016) and may reflect the highest SQT species diversity and strongest discrepancy between simulated and measured emissions that has been observed directly after the rainfall. The SQT emissions may therefore be even stronger in seasons and regions with rapid fungal growth rates.

Under the MM regime,..."

REVIEWERS' COMMENTS:

Reviewer #1 (Remarks to the Author):

Excellent study, congratulations!

Reviewer #4 (Remarks to the Author):

The authors have fully followed reviewers' recommendations and all the points raised have been satisfactorily addressed.

I can recommend now this paper for publication in Nature Communication.